# The Implicit Bias of Stochastic AdaGrad-Norm on Separable Data

## Abstract

This work explores stochastic adaptive gradient descent, i.e., stochastic AdaGrad-Norm, when applied to linearly separable datasets. For the stochastic AdaGrad-Norm method equipped with a wide range of sampling noise, we demonstrate its almost surely convergence result to the $\mathcal{L}^2$ max-margin solution. This means that stochastic AdaGrad-Norm has an implicit bias that yields good generalization, even without regularization terms. We show that the convergence rate of the classification direction is $o(1/\ln^{(1-\epsilon)/2} n)$. Our approach takes a novel stance by explicitly characterizing the $\mathcal{L}^2$ max-margin direction. By doing so, we overcome the challenge that arises from the dependency between the stepsize and the gradient and also address the limitations in the previous AdaGrad-Norm analyses.

## 1 Introduction

With the growth of computing power in recent years, various models like neural networks have gained the ability to perfectly fit training data. These models have a capacity that exceeds the data's capacity, which are referred to as over-parametrized models. Over-parametrized models often exhibit numerous global optimums, yielding a zero training loss, yet exhibiting substantial disparities in test performance (Wu et al., 2018; Chatterji et al., 2022). Fascinatingly, investigations have indicated that optimization algorithms tend to converge towards those optimal points associated with a good generalization (Zhang et al., 2021). This intriguing phenomenon is referred to as the implicit bias of optimizers and is widely speculated to exist (Neyshabur et al., 2014; Zhang et al., 2005; Keskar et al., 2017; Wilson et al., 2017).

Evidence of implicit bias has been established under different settings. For the linear classification task with cross-entropy loss, Soudry et al. (2018) demonstrate that gradient descent (GD) converges to the $\mathcal{L}^2$ max-margin solution. This solution is also commonly known as the hard support vector machine (hard-margin SVM) solution. This revelation underscores that even fundamental optimizers like GD have an implicit bias. Subsequent endeavors have extended their work, adapting GD into stochastic gradient descent (SGD), momentum-based SGD (mSGD), and deterministic adaptive diagonal gradient descent (AdaGrad-Diagonal) (Gunasekar et al. (2018); Qian & Qian (2019); Wang et al. (2021b;a); Wu et al. (2021)). In recent years, as a varient of deterministic AdaGrad-Diagonal, stochastic AdaGrad-Norm has been a major focus in the theoretical community (Faw et al. (2022); Wang et al. (2023); Jin et al. (2022)). The iterates generated by the stochastic AdaGrad-Norm method enjoy the following dynamics (see Streeter & Mcmahan (2010); Ward et al. (2020)):

$$S_n = S_{n-1} + \left\| \nabla g(\theta_n, \xi_n) \right\|^2, \theta_{n+1} = \theta_n - \frac{\alpha_0}{\sqrt{S_n}} \nabla g(\theta_n, \xi_n), \tag{1}$$

where $g(\theta)$ refers to the objective function, $\nabla g(\theta, \xi_n)$ is an unbiased estimation of the gradient $\nabla g(\theta)$ with $\{\xi_n\}$ being mutually independent. $S_n$ is the cumulative stochastic gradient norm, and $\alpha_0 > 0$ represents the constant step size. Compared to traditional GD or SGD, AdaGrad-Norm maintains a uniform update step size for continuously shared stochastic gradient components and eliminates the need for a complex learning rate tuning process. However, to the best of our knowledge, most of these theoretical analyses (Faw et al. (2022); Wang et al. (2023); Jin et al. (2022)) have focused on studying its convergence. A critical question then arises:

*Can stochastic AdaGrad-Norm converge to the $\mathcal{L}^2$ max-margin solution?*

If the answer is true, we can show that stochastic AdaGrad-Norm has an implicit bias.

**Formulation of the convergence** We investigate the linear classification problem with linearly separable data set $\{(x_i, y_i)\}_{i=1}^{N}$, where $y_i \in \{0, 1\}$. The $\mathcal{L}^2$ max-margin solution $\theta^*/\|\theta^*\|$ as the vectors that maximizes the margin between positive data ($y_i = 1$) and negative data ($y_i = 0$), i.e.,

$$\frac{\theta^*}{\|\theta^*\|} := \left\{ \frac{\theta}{\|\theta\|} \ \middle| \ \theta \in \arg\max_{\phi \in \mathbb{R}^d} \min_{1 \leq i \leq N} \left\{ \frac{\mathrm{sgn}(y_i - 0.5)(x_i^\top \phi)}{\|\phi\|} \right\} \right\}, \tag{2}$$

where $\| \cdot \|$ denotes $\ell_2$ norm. Denote the cross-entropy loss $g(\theta) = \frac{1}{N} \sum_{i=1}^{N} g(\theta, x_i)$, where $g(\theta, x_i) = -y_i \ln(\hat{y}_i) - (1 - y_i) \ln(1 - \hat{y}_i)$ and $\hat{y}_i = \frac{1}{1+e^{-\theta^\top x_i}}$. Our main goal is to show that running stochastic AdaGrad-Norm (1) on the cross-entropy loss $g(\theta)$ obtains $\frac{\theta_n}{\|\theta_n\|} \to \frac{\theta^*}{\|\theta^*\|}$ $a.s.$.

For a detailed description of the problem formulation and its background, please refer to Section 2.

**Challenges in analyzing stochastic AdaGrad-Norm** Compared to SGD, mSGD, and deterministic AdaGrad-Diagonal, the analysis of stochastic AdaGrad-Norm presents distinct challenges arising from the following four aspects.

(I) Even if one manages to demonstrate the last-iterate convergence of the objective function $g(\theta_n) \to 0$, it only implies $\theta_n \to \infty$, leaving the limit of the $\mathcal{L}^2$ max-margin direction, i.e., $\theta_n/\|\theta_n\|$, unknown. Since the $\mathcal{L}^2$ max-margin direction is important in some machine learning problems, such as classification, reinforcement learning, we must conduct additional effort to establish convergence of the $\mathcal{L}^2$ max-margin direction. Moreover, the relevant techniques used to prove the last-iterate convergence for stochastic AdaGrad-Norm cannot be directly applied to establish the corresponding results for implicit bias. We will explain why the techniques cannot be transferred in Section 4 after Theorem 4.1.

(II) Previous results on the implicit bias of SGD and mSGD are based on the situation that mini-batch stochastic gradient holds (see Section 3 for more details). Specifically, they use the strong growth property which holds for the mini-batch stochastic gradient, i.e., $\mathbb{E}_{\xi_n} \|\nabla g(\theta, \xi_n)\|^2 \leq M \|\nabla g(\theta)\|^2$. In contrast, the stochastic AdaGrad-Norm method is not related to the choice of mini-batch gradient.

(III) For the stochastic AdaGrad-Norm, the properties of the generated iterate points $\theta_n$ are sensitive to the distance between $\theta_n$ and the stationary point. Such a challenge does not exist in previous settings. For example, considering deterministic or stochastic algorithms under a quadratic growth condition, this challenge is successfully bypassed by considering the dynamic system in different segments. However, for the stochastic AdaGrad-Norm, the segment of iterates near and far from the stationary point is highly random, making the previous technique unavailable. Therefore, it becomes challenging in this setting,

**Related works** There are several works that is related to this topic. For example, Soudry et al. (2018) prove that GD converges to the $\mathcal{L}^2$ max-margin solution for linear classification tasks with exponential-tailed loss. In Nacson et al. (2019), authors mainly extended the form of the loss from the work of Soudry et al. (2018), and considered the case of linear neural networks. For SGD and momentum-based SGD, Wang et al. (2021a) prove the convergence to the $\mathcal{L}^2$ max-margin solution for linear classification task with exponential-tailed loss and mini-batch stochastic gradient.

For deterministic AdaGrad-Diagonal, Soudry et al. (2018); Gunasekar et al. (2018); Qian & Qian (2019) claim that it does not converge to the $\mathcal{L}^2$ max-margin solution as the non-adaptive methods do (e.g. SGD, GD). Instead, for stochastic AdaGrad-Norm, Jin et al. (2022) presents the last-iterate convergence. Wang et al. (2023) and Faw et al. (2022) obtained the convergence rates of stochastic AdaGrad-Norm. The characterization of the converging point (like implicit bias) of stochastic AdaGrad-Norm remains unknown.

**Contributions** In this work, we present an affirmative response to the aforementioned question. Specifically, we provide rigorous proof demonstrating the almost surely convergence of the stochastic AdaGrad-Norm method to the $\mathcal{L}^2$ max-margin solution. This result emphasizes that the resulting classification hyperplane closely conforms to the solution obtained through the application of the hard support vector machine (see Theorems 4.2 and 4.3).

In comparison to previous works that mainly focused on mini-batch stochastic gradient (Wang et al., 2021b), our study stands out by its capacity to handle a wide range of stochastic settings (Assumption

3.1). As a special case, our study can be applied to any stochastic algorithms with bounded noise, i.e., $\nabla g(\theta, \xi_n) = \nabla g(\theta) + \xi_n$, (for some $\xi_n, \sup_{n \geq 1} \|\xi_n\| < +\infty$), and the stochastic algorithms with mini-batch stochastic gradient.

We conducted a small numerical experiment in Section 5 to further support our conclusions.

Our technical contributions are summarized as follows:

(I) We begin by adopting a divide-and-conquer approach, simultaneously applying a specific indicator function at both ends of the stochastic dynamical system. This new approach allows us to analyze the generated iterate points' properties properly. When the iterate point is close to the stationary point, we leverage second-order information from the loss function to provide a deeper characterization of the algorithm's behavior. Conversely, when the iterate point is far from the stationary point, we establish a local strong growth property. Combining these two scenarios, and exploiting the separability property inherent in the dataset, we conclude that the AdaGrad-Norm algorithm converges towards a max-margin solution.

(II) In a separate line of investigation, we employ the martingale method to establish the almost everywhere convergence result. This outcome enables us to convert the convergence order of the partition vector into an order related to the iterates' norm, specifically, $\left\| \frac{\theta_n}{\|\theta_n\|} - \frac{\theta^*}{\|\theta^*\|} \right\|^2 = O(\|\theta_n\|^{-\alpha})$ $(\forall\, 0 < \alpha < 1)$ $a.s..$ By combining this result with the earlier amplitude findings, we eventually derive the convergence rate of the partition vector as $\min_{1 \leq k \leq n} \left\| \frac{\theta_k}{\|\theta_k\|} - \frac{\theta^*}{\|\theta^*\|} \right\| = o\left( \ln^{-\frac{1-\epsilon}{2}} n \right)$ $(\forall\, \epsilon > 0)$ $a.s..$

## 2 PROBLEM FORMULATION

In this section, we give a detailed formulation of our aimed problem. We consider the linear classification problem with linearly separable data set $\{(x_i, y_i)\}_{i=1}^N$, where $y_i \in \{0, 1\}$. Here, separability means that there exists a vector $\theta_0 \in \mathbb{R}^d$, such that for any $y_i = 1$, $\theta_0^\top x_i > 0$, and for any $y_i = 0$, $\theta_0^\top x_i < 0$. Meanwhile, we call $\theta_0$ as a margin vector. The setting has been considered in many existing works (Soudry et al. (2018); Wang et al. (2021a); Qian & Qian (2019)).

Define a $\sigma$-filtration $\mathscr{F}_n := \sigma\{\theta_1, \xi_1, \xi_2, \ldots, \xi_{n-1}\}$. Denote $\|\cdot\|$ as the $\ell_2$ norm. Denote the $\mathcal{L}^2$ max-margin solution as $\theta^* / \|\theta^*\|$, which can be formulation by Equation (2). It is also common in the literature to denote

$$\frac{\theta^*}{\|\theta^*\|} := \left\{ \frac{\theta}{\|\theta\|} \,\middle|\, \theta \in \arg\min_{\phi \in \mathbb{R}^d} \left\{ \|\phi\| \,\middle|\, \mathrm{sgn}(y_i - 0.5)(\phi^\top x_i) \geq 1, \, \forall i \right\} \right\}.$$

The two definitions are equivalent.

**Remark 1.** *The existing techniques of the works (Soudry et al., 2018; Wang et al., 2021a; Qian & Qian, 2019) are all based on the dual definition. In contrast to their techniques, our analysis is completely based on the original definition (Equation (2)) for the proof.*

In this paper, we utilize the cross-entropy loss function along with the linear sigmoid activation function, i.e., $g(\theta) = \frac{1}{N} \sum_{i=1}^N g(\theta, x_i)$, where $g(\theta, x_i) = -y_i \ln(\hat{y}_i) - (1 - y_i) \ln(1 - \hat{y}_i)$ and $\hat{y}_i = \frac{1}{1 + e^{-\theta^\top x_i}}$. This setup is indeed the familiar logistic regression. This is a special case of the exponential-tail loss, as discussed in Soudry et al. (2018); Wang et al. (2021a). Since the choice of logistic regression does not affect the validity of our analysis, while the use of exponential-tail loss does introduce many tedious notations, we present our results under the logistic regression setting in the rest of this paper for brevity. Our results can easily be generalized to the stochastic AdaGrad-Norm method with tight exponential-tail loss.[1] For function $g$, we have the following property.

**Property 1.** *The gradient of the loss function, denoted as $\nabla g(\theta)$, satisfies Lipschitz continuity, i.e., $\forall\, \theta_1,\, \theta_2 \in \mathbb{R}^d$, there is $\|\nabla g(\theta_1) - \nabla g(\theta_2)\| \leq c\|\theta_1 - \theta_2\|$, where $c$ is the Lipschitz constant of the function $\nabla g(\theta)$.*

---

[1]We will justify this generalization in Appendix C.11.

Due to the particularity of classification problems, the global optimal point does not exist. When $\theta_n$ tends to infinity along a certain margin vector, the value of the loss function tends to zero. For any $\epsilon > 0$ and any margin vector $e$, there exists a positive constant $N_0$ associated with $e$, such that for any $\theta/\|\theta\| = e$ and $\|\theta\| > N_0$, we have $g(\theta) < \epsilon$, i.e.,

$$\lim_{\|\theta\| \to +\infty, \theta/\|\theta\| = e} g(\theta) = 0,$$

where $e$ is a margin vector of the data set $\{(x_i, y_i)\}_{i=1}^N$. However, we are more interested in the direction $e$ that the regression vector $\theta/\|\theta\|$ converges to. We aim to show that this vector converges to the direction of the L2 max-margin of the hard-margin SVM.

In the following, we will give the convergence results of the stochastic AdaGrad-Norm method, described in (1), with the aforementioned objective function $g(\theta)$.

## 3 STOCHASTIC GRADIENT SETTING

The results we are going to present hold for the natural noise model induced by mini-batch sampling. Nevertheless, to incorporate a broader family of noise model, such as the bounded variance model, we present a general noise model under which we derive our main results.

We first give our assumption on the unbiased estimation $\nabla g(\theta, \xi_n)$ of the gradient. Here, unbiasedness implies that $\mathbb{E}_{\xi_n} \nabla g(\theta, \xi_n) = \nabla g(\theta)$.

**Assumption 3.1.** *There exist $M_0 > 0$, $a > 0$, such that the variance of $\nabla g(\theta, \xi_n)$ satisfies*

$$\mathbb{E}_{\xi_n} \left\| \nabla g(\theta, \xi_n) \right\|^2 \leq M_0 \left\| \nabla g(\theta) \right\|^2 + a.$$

*Meanwhile, there exist $\delta_0 > 0$, $\hat{K} > 0$, such that when $g(\theta) < \delta_0$, there is $\|\nabla g(\theta, \xi_n)\| \leq \hat{K}$ a.s..*

It is worth remarking that Assumption 3.1 differs from that in the existing works on the implicit bias of stochastic algorithms, in which mini-batch stochastic gradient is taken into consideration. In contrast, we consider all estimation noise in the assumption, which includes the mini-batch stochastic gradient (see the following elaboration).

**Mini-batch stochastic gradient** The mini-batch stochastic gradient is given by

$$\nabla g(\theta, \xi_n) = \frac{1}{|C_{i_n}|} \sum_{\bar{x} \in C_{i_n}} \nabla g(\theta, \bar{x}), \tag{3}$$

where $C_{i_n}$ is a randomly selected mini-batch from the given data set. Through Lemma 8 in Wang et al. (2021b), we know that sampling noise satisfies the *strong growth condition*, i.e., $\mathbb{E}_{\xi_n} \|\nabla g(\theta, \xi_n)\|^2 \leq \tilde{M} \|\nabla g(\theta)\|^2$.

Since any subset (mini-batch) of a linearly separable data set is separable, we know that $\theta$ satisfying $g(\theta) < \delta_0$ is a margin vector of $\{x_i, y_i\}$ by Lemma B.10 with $\delta_0 = (\ln 2)/2N$. Then by Lemma B.8, we have

$$\|\nabla g(\theta, \xi_n)\| = \frac{1}{|C_{i_n}|} \left\| \sum_{\bar{x} \in C_{i_n}} \nabla g(\theta, \bar{x}) \right\| \leq \frac{1}{|C_{i_n}|} \sum_{\bar{x} \in C_{i_n}} \|\nabla g(\theta, \bar{x})\| \leq \frac{k_2}{|C_{i_n}|} \sum_{\bar{x} \in C_i} g(\theta, \bar{x})$$

$$\leq \frac{k_2 N}{|C_{i_n}|} g(\theta) < \frac{k_2}{|C_{i_n}|} \cdot \frac{\ln 2}{2} =: \hat{K}.$$

Hence the mini-batch stochastic gradient satisfies Assumption 3.1. Our setting also encompasses more stochastic gradients. In reinforcement learning, the policy we aim to learn can be viewed as a classification problem in the policy space. However, instead of simple mini-batch sampling, we perform Markov sampling based on the parameterized policy for policy gradients. Consequently, the policy gradients in reinforcement learning generally do not satisfy the strong growth condition (even when the parameterized policy is an over-parameterized model). Nonetheless, since the rewards obtained at each step in the reinforcement learning setting are bounded, it can be shown that the stochastic gradients are almost surely bounded. Therefore, they still fit within our setting.

## 4 MAIN RESULTS

Now, we are ready to present our main results. We first recall the last-iterate convergence result of stochastic AdaGrad-Norm, which was previously proved by Jin et al. (2022).

**Theorem 4.1.** *(Theorem 3 in Jin et al. (2022)) Suppose that Assumption 3.1 holds. Consider the classification problem with the cross-entropy loss on a linearly separable data set (Section 2). For the stochastic AdaGrad-Norm method given in Equation (1) equipped with step size $\alpha_0 > 0$ and initial parameter $\theta_1 \in \mathbb{R}^d$, we have $g(\theta_n) \to 0$ a.s., and $\|\theta_n\| \to +\infty$ a.s..*

The original proof of this theorem can be found in the referred paper, but to make this manuscript self-contained, we provide a proof of this theorem with our notation and convention in Appendix C.8. Nevertheless, the method in the original proof cannot be directly applied to the analysis for the implicit bias. The original proof constructs a recursive iterative inequity therein for $g(\theta)$, i.e.,

$$g(\theta_{n+1}) - g(\theta_n) \leq \frac{k}{S_{n-1}} + c_n \tag{4}$$

with $\sum_{n=1}^{+\infty} c_n < +\infty$ and $k > 0$. Then, their goal is to prove that the difference between $\|\nabla g(\theta_{n+1})\|^2$ and $\|\nabla g(\theta_n)\|^2$ becoming sufficiently small as the iterations progress. To do so, they try to bound $\|\nabla g(\theta_{n+1})\|^2 - \|\nabla g(\theta_n)\|^2$ via $g(\theta_{n+1}) - g(\theta_n)$ and inequity $\|\nabla g(\theta)\|^2 \leq 2cg(\theta)$ for Lipschitz constant $c$ of $\nabla g$. However, to obtain the implicit bias, their techniques become unsuitable due to the nuanced nature of Lyapunov function, i.e., $\|\theta_n/\|\theta_n\| - \theta^*/\|\theta^*\|\|^2$. Specifically, the terms $\nabla(\|\theta_n/\|\theta_n\| - \theta^*/\|\theta^*\|\|^2)^\top \nabla g(\theta_n, \xi_n)/\sqrt{S_n}$ and $\|\theta_n/\|\theta_n\| - \theta^*/\|\theta^*\|\|^2$ lack a clear and evident quantitative relationship, making it difficult to obtain Equation (4). Consequently, novel methods and techniques become imperative to address this challenge.

Next, we present the almost surely convergence analysis of the $\mathcal{L}^2$ max-margin direction $\theta_n/\|\theta_n\|$.

**Theorem 4.2.** *Suppose that Assumption 3.1 holds. Consider the classification problem with the cross-entropy loss on a linearly separable data set (Section 2). For the stochastic AdaGrad-Norm method given in Equation (1) equipped with step size $\alpha_0 > 0$ and initial parameter $\theta_1 \in \mathbb{R}^d$, we have*

$$\frac{\theta_n}{\|\theta_n\|} \to \frac{\theta^*}{\|\theta^*\|} \quad a.s.,$$

*where $\theta^*/\|\theta^*\|$ is the $\mathcal{L}^2$ max-margin solution.*

In Theorem 4.2, we prove that the stochastic AdaGrad-Norm method has the implicit bias to find the $\mathcal{L}^2$ max-margin solution.

Below, we provide the intuition behind the proof.

**Intuition of the theorem**  We know the update direction of $\theta_n$ under stochastic AdaGrad-Norm can be seen as $-\nabla g(\theta_n, \xi_n)$, which is an unbiased estimate of the reverse gradient $-\nabla g(\theta_n)$. By virtue of *the law of large numbers*, we can roughly think that the update direction of $\{\theta_n\}$ is $-\nabla g(\theta_n)$ when $n \to +\infty$ (we will need additional arguments). We know if this direction is a reverse direction of $\nabla(\|\theta_n/\|\theta_n\| - \theta^*/\|\theta^*\|\|^2)$, this algorithm has the power to make $\|\theta_n/\|\theta_n\| - \theta^*/\|\theta^*\|\|^2$ close to 0. This means, we just need to inspect the value of the term $-\nabla g(\theta_n)^T \nabla(\|\theta_n/\|\theta_n\| - \theta^*/\|\theta^*\|\|^2)$. When we expand the term we will obtain

$$-\nabla g(\theta_n)^\top \nabla \left\| \frac{\theta_n}{\|\theta_n\|} - \frac{\theta^*}{\|\theta^*\|} \right\|^2 \approx \frac{f_{x_i}(\theta_n, x_{i_n})\|\theta_n\|}{N(\|\theta_n + 1\|)^2 \sqrt{S_{n-1}}} \left( \sum_{i \in \mathbf{i}_n} \psi_i \left( \frac{\theta_n^\top x_i}{\|\theta_n\|} - \hat{\theta}^{*\top} x_i \right) \right.$$

$$+ \hat{k}_0 \sum_{i \notin \mathbf{i}_n}^N \frac{\psi_i}{e^{(d_{n,i} - d_{n,i_n})(\|\theta_n\| + 1)}} \left( \frac{\theta_n^\top x_i}{\|\theta_n\|} - \hat{\theta}^{*\top} x_i \right) \right)$$

$$:= A_n + B_n,$$

where $\psi_i := \text{sgn}(y_i - 0.5)$, $\mathbf{i}_n := \{i | i = \arg\min_{1 \leq i \leq N} \psi_i \theta_n^\top x_i / \|\theta_n\|\}$, $d_{n,i} := |\theta_n^\top x_i| / \|\theta_n\|$ and $i_n$ is a element of $\mathbf{i}_n$. We can find that $x_{i_n}$, which is the closest vector to $\theta_n$, plays a leading role.

In the above inequality, $A_n$ is the negative term, and it is bigger in absolute value (see Step 2 of the proof sketch for the reason). That concludes $A_n$ is the dominate term, so we can obtain $-\nabla g(\theta_n)^T \nabla(\|\theta_n/\|\theta_n\| - \theta^*/\|\theta^*\|\|^2) < 0$ as desired. That means stochastic AdaGrad-Norm has a tendency to make $\|\theta_n/\|\theta_n\| - \theta^*/\|\theta^*\|\|^2$ decrease as $n$ increases.

Notice that this is just an intuitive understanding. The full proof will be quite long and we defer that to Appendix C.9. Now we provide a proof sketch that illustrates the main arguments that compose the proof.

*Proof Sketch.* Given

$$f(\theta) := 1 - \frac{\theta^\top \hat{\theta}^*}{\|\theta\| + 1}$$

with $\hat{\theta}^* := \theta^*/\|\theta^*\|$, which tends to $\left\|\frac{\theta}{\|\theta\|} - \frac{\theta^*}{\|\theta^*\|}\right\|^2$ as $\|\theta\| \to +\infty$. We now aim to prove $f(\theta_n) \to 0$ *a.s.*.

**Step 1**: In this step, we construct a recursive inequality for $f(\theta_n)$. We derive that

$$\mathbb{E}\left(f(\theta_{n+1})\right) - \mathbb{E}\left(f(\theta_n)\right) \le -\mathbb{E}\left(\left(\frac{\hat{\theta}^*\|\theta_n\| - \frac{\theta_n \theta_n^\top \hat{\theta}^*}{\|\theta_n\|}}{(\|\theta_n\| + 1)^2}\right)^\top \frac{\alpha_0 \nabla g(\theta_n)}{\sqrt{S_{n-1}}}\right) + \mathbb{E}\left(G_n\right), \quad (5)$$

where

$$G_n := \left|\left(\frac{\hat{\theta}^*\|\theta_n\| - \frac{\theta_n \theta_n^\top \hat{\theta}^*}{\|\theta_n\|}}{(\|\theta_n\| + 1)^2}\right)^\top \alpha_0 \nabla g(\theta_n, \xi_n)\left(\frac{1}{\sqrt{S_{n-1}}} - \frac{1}{\sqrt{S_n}}\right)\right| + \frac{T_n \alpha_0^2 \|\nabla g(\theta_n, \xi_n)\|^2}{S_n}$$

$$+ \frac{\alpha_0 \hat{\theta}^{*\top} \nabla g(\theta_n, \xi_n)}{(\|\theta_n\| + 1)^2 \sqrt{S_{n-1}}} + \frac{N^2 \max_{1 \le i \le N}\{\|x_i\|^2\}}{2k_1^2 \ln^2 2} \cdot \frac{\|\nabla g(\theta_n)\|^2}{\sqrt{S_{n-1}}},$$

where $T_n$ is defined in Equation (63). It can be shown that $\sum_{n=1}^{+\infty} \mathbb{E}(G_n) < +\infty$ (see the specific proof in Appendix C.9). Thus, we focus on studying the first term on the right-hand side of Equation (5).

**Step 2** In this step, we focus on decomposing the first term in Equation (5).

$$\mathbb{E}\left(\left(\frac{\hat{\theta}^*\|\theta_n\| - \frac{\theta_n \theta_n^\top \hat{\theta}^*}{\|\theta_n\|}}{(\|\theta_n\| + 1)^2}\right)^\top \frac{\nabla g(\theta_n)}{\sqrt{S_{n-1}}}\right) \le \mathbb{E}\left(\frac{1}{N\sqrt{S_{n-1}}}\sum_{i=1}^N \psi_i \frac{\theta_n^\top x_i - \hat{\theta}^{*\top} x_i \|\theta_n\|}{(\|\theta_n\| + 1)^2}\right) := \mathbb{E}(H_n),$$

where the definition of $f_{x_i}(\theta, x_i)$ can refer Equation (38) in Appendix C.9 and $\psi_i := \text{sgn}(y_i - 0.5)$. We then prove that the right-hand side of the above inequality is negative. Denote the index of the support vector as $\mathbf{i}_n := \{i | i = \arg\min_{1 \le i \le N} \psi_i \theta_n^\top x_i/\|\theta_n\|\}$, and $i_n$ is a element of $\mathbf{i}_n$. Then we have $\exists \hat{k}_0 > 0$, such that

$$H_n \le \frac{f_{x_i}(\theta_n, x_{i_n})\|\theta_n\|}{N(\|\theta_n + 1\|)^2 \sqrt{S_{n-1}}}\left(\sum_{i \in \mathbf{i}_n} \psi_i \left(\frac{\theta_n^\top x_i}{\|\theta_n\|} - \hat{\theta}^{*\top} x_i\right)\right.$$

$$\left. + \hat{k}_0 \sum_{i \notin \mathbf{i}_n}^N \frac{\psi_i}{e^{(d_{n,i} - d_{n,i_n})(\|\theta_n\| + 1)}}\left(\frac{\theta_n^\top x_i}{\|\theta_n\|} - \hat{\theta}^{*\top} x_i\right)\right) := A_n + B_n, \quad (6)$$

where $d_{n,i} := |\theta_n^\top x_i|/\|\theta_n\|$. We can assert that the first term $A_n$ on the right side of the inequality is negative. But at this moment, we cannot obtain $\sum_{n=1}^{+\infty} \mathbb{E}|B_n| < +\infty$. This requires the introduction of $\mathbf{C}_n^+$ and $\mathbf{C}_n^-$. We continue to answer specific questions about these two sets in the next step.

**Step 3**: From a high-level perspective, since we cannot confirm that $\sum_{n=1}^{+\infty} \mathbb{E}|B_n| < +\infty$, we need to further decompose $B_n$. We wish to achieve a decomposition $B_n = C_n + D_n$, where we allow $C_n > 0$ but not excessively large; specifically, it must satisfy $C_n < \frac{-A_n}{2}$, and at the same time, we have $\sum_{n=1}^{+\infty} \mathbb{E}|D_n| < +\infty$. By merging these two decompositions, we obtain $H_n \le \frac{A_n}{2} + D_n$, which meets our need. When completing the second decomposition $B_n = C_n + D_n$, we need to introduce

the sets $\mathcal{C}_n^-$ and $\mathcal{C}_n^+$, where $\mathcal{C}_n^+ := \{\|(\theta_n/\|\theta_n\|) - \hat{\theta}^*\| \geq \mathcal{L}\}$, $\mathcal{C}_n^- := \{\|(\theta_n/\|\theta_n\|) - \hat{\theta}^*\| < \mathcal{L}\}$. In these two different sets, our ways of decomposition are different. In the case where $\mathcal{C}_n^-$ occurs, that is, when $\theta/\|\theta\|$ is to $\hat{\theta}^*$, we have the following geometric relationship lemma.

**Lemma 4.1.** *Let $\{x_i\}_{i=1}^N$ be d-dimensional vectors. Then there is a vector $x_\theta$ such that $|\theta^\top x_\theta|/\|\theta\| := \min_{1 \leq i \leq N} \{|\theta^\top x_i|/\|\theta\|\}$. Let $\theta^*/\|\theta^*\|$ as the max-margin vector. Then there exists $\delta_0 > 0$, $\hat{r} > 0$, such that for all $\theta/\|\theta\| \in U(\theta^*/\|\theta^*\|, \delta_0)/\{\theta^*/\|\theta^*\|\}$, where $U(\theta^*/\|\theta^*\|, \delta_0)$ means $\delta_0$-neighborhood of vector $\theta^*/\|\theta^*\|$, it holds $\left|\frac{\theta^\top x_i}{\|\theta\|} - \frac{\theta^* x_i}{\|\theta^*\|}\right| < \hat{r} \left|\frac{\theta^\top x_\theta}{\|\theta\|} - \frac{\theta^* x_\theta}{\|\theta^*\|}\right|$ $(\forall\, i \in [1, N])$.*

Through this lemma, we obtain

$$\sum_{i \notin \mathbf{i}_n} \mathbf{1}_{\mathcal{C}_n^-} \frac{\psi_i}{e^{(d_{n,i} - d_{n,i_n})(\|\theta_n\|+1)}} \left(\frac{\theta_n^\top x_i}{\|\theta_n\|} - \hat{\theta}^{*\top} x_i\right) \leq \hat{k}_0 \hat{c} N \frac{\hat{U}}{\|\theta_n\|+1} + \hat{k}_0 \frac{N\hat{r}}{e^{\hat{U}}} \left|\frac{\theta_n^\top x_{i_n}}{\|\theta_n\|} - \hat{\theta}^{*\top} x_{i_n}\right|,$$

where $\hat{U}$ is an undetermined constant. Similarly, where $\mathcal{C}_n^+$ occurs, we get

$$\sum_{i \notin \mathbf{i}_n} \mathbf{1}_{\mathcal{C}_n^+} \frac{\psi_i}{e^{(d_{n,i} - d_{n,i_n})(\|\theta_n\|+1)}} \left(\frac{\theta_n^\top x_i}{\|\theta_n\|} - \hat{\theta}^{*\top} x_i\right) \leq \frac{N \cdot \tilde{M}_1}{e^{s'\|\theta_n\|}} + \hat{k}_1 \frac{N}{e^{\hat{U}}} \left|\frac{\theta_n^\top x_{i_n}}{\|\theta_n\|} - \hat{\theta}^{*\top} x_{i_n}\right|,$$

where $M_1$ is a constant. Combining the arguments, we get

$$\sum_{i \notin \mathbf{i}_n} \frac{\psi_i}{e^{(d_{n,i} - d_{n,i_n})(\|\theta_n\|+1)}} \left(\frac{\theta_n^\top x_i}{\|\theta_n\|} - \hat{\theta}^{*\top} x_i\right) \leq (\hat{k}_0 \hat{r} + \hat{k}_1) \frac{N}{e^{\hat{U}}} \left|\frac{\theta_n^\top x_{i_n}}{\|\theta_n\|} - \hat{\theta}^{*\top} x_{i_n}\right| + \frac{N \cdot \tilde{M}_1}{e^{s'\|\theta_n\|}}$$

$$+ \hat{k}_0 \hat{c} N \frac{\hat{U}}{\|\theta_n\|+1}.$$

By adjusting the value of $\hat{U}$, we can always cancel out the first term with the half of the negative term in Equation (6), and then we only need to prove that the remainder term can be neglected. That is to prove

$$\sum_{n=1}^{+\infty} \mathbb{E}\left(\frac{f_{x_i}(\theta_n, x_{i_n})\|\theta_n\|}{N(\|\theta_n + 1\|)^2 \sqrt{S_{n-1}}} \cdot \left(\frac{N \cdot \tilde{M}_1}{e^{s'\|\theta_n\|}} + \hat{k}_0 \hat{c} N \frac{\hat{U}}{\|\theta_n\|+1}\right)\right) < +\infty.$$

**Step 4** In this step, we will prove the convergence of the series sum in the final step of the third step. We prove this conclusion by the following lemma:

**Lemma 4.2.** *Consider the AdaGrad-Norm Equation (1) under our problem setting in Section 2 and Assumption 3.1. We have for any $\alpha_0 > 0$, $\alpha > 0$, $\theta_1$, there is $\sum_{k=2}^n \mathbb{E}\left(\frac{\|\nabla g(\theta_k)\|^2}{\sqrt{S_{k-1} g(\theta_k) \ln^{1+\alpha}(g(\theta_k))}}\right) < +\infty$.*

**Step 5** Through the above steps, we have obtained the following recursive formula:

$$\mathbb{E}(f(\theta_{n+1}|\mathscr{F}_n) - f(\theta_n) \leq -\frac{1}{2} \frac{f_{x_i}(\theta_n, x_{i_n})\|\theta_n\|}{N(\|\theta_n + 1\|)^2 \sqrt{S_{n-1}}} \sum_{i \in \mathbf{i}_n} \psi_i \left(\frac{\theta_n^\top x_i}{\|\theta_n\|} - \hat{\theta}^{*\top} x_i\right) + c_n,$$

where $\sum_{n=1}^{+\infty} c_n < +\infty$. According to the martingale difference sum convergence theorem, we can conclude that $f(\theta_n)$ convergence almost surely. Then, we prove by contradiction that this limit can only be 0. We assume that this limit is not 0, and immediately derive a contradiction from the following result:

$$\sum_{n=2}^{+\infty} \frac{\|\theta_n\| f_{x_{i_n}}(\theta_n, x_{i_n})}{N(\|\theta_n\|+1)^2 \sqrt{S_{n-1}}} > q_1 \sum_{n=1}^{+\infty} \left(\ln\|\theta_{n+1}\| - \ln\|\theta_n\|\right) - q_2 \sum_{n=1}^{+\infty} \frac{\|\nabla g(\theta_n, \xi_n)\|^2}{\|\theta_n\|^2 S_n} = +\infty \ \ a.s..$$

Therefore, we have proved this theorem. $\qquad\square$

Clearly, we can directly derive the following corollary regarding mini-batch stochastic gradients from our theorem.

**Corollary 4.1.** *Suppose that mini-batch stochastic gradient holds (Equation (3)). Consider the classification problem with the cross-entropy loss on a linearly separable data set (Section 2). For the stochastic AdaGrad-Norm method given in Equation (1) equipped with step size $\alpha_0 > 0$ and initial parameter $\theta_1 \in \mathbb{R}^d$, we have*

$$\frac{\theta_n}{\|\theta_n\|} \to \frac{\theta^*}{\|\theta^*\|} \;\; a.s.,$$

*where $\theta^*/\|\theta^*\|$ is the $\mathcal{L}^2$ max-margin solution.*

Previous works (Soudry et al., 2018; Gunasekar et al., 2018; Qian & Qian, 2019) have pointed out that whether AdaGrad-Diagonal converges to the max-margin direction depends on the initial point and step size depends on the initial point and step size. They subsequently concluded that it is not as predictable and robust as the non-adaptive methods (e.g., SGD, GD). However, the claim only holds for the deterministic AdaGrad-Diagonal method, which is described by the system

$$\theta_{n+1} = \theta_n - \eta \mathbf{G}_n^{-1/2} \nabla g(\theta_n),$$

where $\mathbf{G}_n \in \mathbb{R}^{d \times d}$ is a diagonal matrix such that, $\forall i: \; \mathbf{G}_n[i,i] = \sum_{k=0}^{n} (\nabla g(\theta_k)[i])^2$. Nonetheless, it is crucial to emphasize the substantial distinctions inherent in the properties of the algorithm under discussion when compared to the stochastic AdaGrad-Norm method. Specifically, the stochastic AdaGrad-Norm method maintains a uniform step size consistently across all components, leading to fundamental differences in the analytical methods and techniques that are used to prove the convergence of these two algorithms. For the AdaGrad-Diagonal algorithm, we can compute the key component, denoted as $-\nabla f(\theta_n)^\top (\theta_{n+1} - \theta_n)$, which determines the update direction of the decision boundary, as analogous to Equation (39). This computation yields the following equation.

$$\mathbb{E}(\nabla f(\theta_n)^\top G_n^{-\frac{1}{2}} \nabla g(\theta_n)) = \mathbb{E}\left( \frac{1}{N\sqrt{S_{n-1}}} \sum_{i=1}^N \mathrm{sgn}(y_i - 0.5) f_{x_i}(\theta_n, x_i) \right.$$
$$\left. \cdot \left( \frac{\theta_n^\top G_n^{-\frac{1}{2}} x_i - \hat{\theta}^{*\top} G_n^{-\frac{1}{2}} x_i \|\theta_n\|}{(\|\theta_n\| + 1)^2} - \frac{\theta_n^\top G_n^{-\frac{1}{2}} x_i}{2(\|\theta_n\| + 1)^2} \left\| \frac{\theta_n}{\|\theta_n\|} - \hat{\theta}^* \right\|^2 \right) \right).$$

Here we have omitted higher-order terms, while the full proof is available in the appendix. It is worth noting that, given the diagonal matrix structure of $G_n$ with distinct diagonal elements, as the iterations progress, our pursuit effectively converges towards identifying the max-margin vector associated with the dataset $\{G_\infty^{-\frac{1}{2}} \cdot x_i, y_i\}_{i=1}^N$. This differs from the previous result.

Finally, we present the convergence rate analysis of the stochastic AdaGrad-Norm method, as shown in Theorem 4.3.

**Theorem 4.3.** *Suppose that Assumption 3.1 holds. Consider the classification problem with the cross-entropy loss on a linearly separable data set (Section 2). For the stochastic AdaGrad-Norm method given in Equation (1) equipped with step size $\alpha_0 > 0$ and initial parameter $\theta_1 \in \mathbb{R}^d$, we have*

$$\min_{1 \le k \le n} \left\| \theta_k/\|\theta_k\| - \theta^*/\|\theta^*\| \right\| = o\left(1/\ln^{\frac{1-\epsilon}{2}} n\right) \;\; (\forall\, 0 < \epsilon < 1)\, a.s.,$$

*where $\theta^*/\|\theta^*\|$ is the $\mathcal{L}^2$ max-margin solution.*

This theorem presents the convergence rate $o\left(1/\ln^{\frac{1-\epsilon}{2}} n\right) \;\; \forall\, \epsilon > 0 \;\; a.s.$ of the $\mathcal{L}^2$ max-margin direction. This result is new to the area.

Analysis against corresponding GD results, given by Soudry et al. (2018), reveals that the convergence rate for both $g(\theta_n)$ and $\theta_n/\|\theta_n\|$ within stochastic AdaGrad-Norm is comparatively slower. This observation is not surprising, as the stochastic AdaGrad-Norm method uses a decreasing step size, which will be much smaller than that used in GD as iteration grows. However, for GD, one has to verify whether the step size $\alpha$ satisfies $\alpha < 2\beta^{-1}\sigma_{\max}^{-2}(X)$ Soudry et al. (2018), where $X$ is the data matrix, $\sigma_{\max}(\cdot)$ denotes the maximal singular value and $\beta$ is a constant characterized by loss function $g$. This checking rule requires an extra burden of hyperparameter tuning. In contrast, the stochastic AdaGrad-Norm method uses simple step sizes.

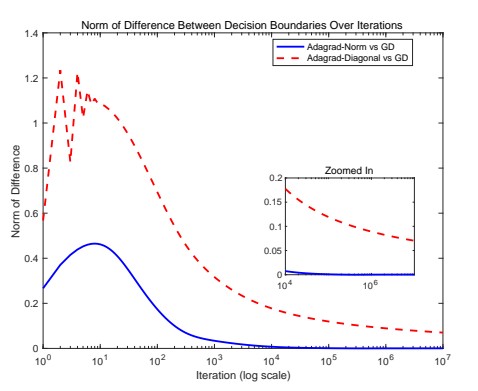
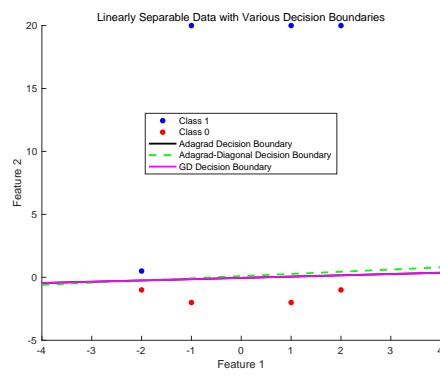

(a) Norm of difference between hyperplanes and SVM solution over iterations

(b) Decision boundary

Figure 1: Numerical experiments on AdaGrad-Norm and other gradient descent variants.

The proof strategy of this theorem is very similar to that of Theorem 4.2. We only need to replace the function $f(\theta)$ in the proof of Theorem 4.2 with $\|\theta\|^\alpha \cdot f(\theta)$ ($\forall\, 0 < \alpha < 1$). We provide the proof in Appendix C.10. We immediately derive the following corollary regarding mini-batch stochastic gradients from our theorem.

**Corollary 4.2.** *Suppose that mini-batch stochastic gradient holds (Equation (3)). Consider the classification problem with the cross-entropy loss on a linearly separable data set (Section 2). For the stochastic AdaGrad-Norm method given in Equation (1) equipped with step size $\alpha_0 > 0$ and initial parameter $\theta_1 \in \mathbb{R}^d$, we have*

$$\min_{1 \le k \le n} \left\| \theta_k/\|\theta_k\| - \theta^*/\|\theta^*\| \right\| = o\big(1/\ln^{\frac{1-\epsilon}{2}} n\big) \ (\forall\, 0 < \epsilon < 1) \ a.s.,$$

*where $\theta^*/\|\theta^*\|$ is the $\mathcal{L}^2$ max-margin solution.*

## 5 EXPERIMENT

On a small dataset containing 8 data points, we run the number of training steps to $10^7$. The Norm of differences between the obtained hyperplanes and the hard SVM solutions is plotted in Figure 1. It is more evident on this dataset that, as iterations progress, the hyperplanes trained by AdaGrad-Norm and gradient descent (GD) tend to overlap, whereas the hyperplanes trained by AdaGrad-Diagonal and GD exhibit a bias that does not converge to zero when the number of step approaches very large. This validates our theoretical findings that AdaGrad-Norm has an implicit bias that is similar to GD, while methods like AdaGrad-Diagonal do not have such properties.

## 6 CONCLUSION

This work focuses on the convergence analysis of the stochastic AdaGrad-Norm method, a variant of the AdaGrad method, with linearly separable data sets. While previous perspectives often suggest that AdaGrad's convergence might hinge on initialization and step size, our findings present a contrasting view. Specifically, we establish that stochastic AdaGrad-Norm exhibits an implicit bias, consistently converging towards the $\mathcal{L}^2$ max-margin solution, even without regularization terms. Furthermore, we present the convergence rates for the $\mathcal{L}^2$ max-margin solution, offering comprehensive insights into the algorithm's convergence dynamics.

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

## A  IMPACT STATEMENTS

This paper presents work whose goal is to advance the field of machine learning and optimization methods. We do not see immediate societal impact. There are potential indirect societal consequences of our work, but we do not foresee them at the moment.

From the technical standpoint, our study overcomes a key challenge arising from the dependency between step size and gradient. The challenge is solved by intricately examining the direction recursion of the optimization variable, enabling us to disentangle these dependencies. Moreover, we tackle extra radial noise through insightful geometric transformations. By harnessing a blend of martingale theory and geometric reasoning, we provide a new way to show the convergence of the $\mathcal{L}^2$ max-margin direction. Notably, the techniques developed in this study extend beyond the boundaries of the stochastic AdaGrad-Norm method. The techniques are also valuable in analyzing the convergence behavior of other stochastic methods, such as Adam, and in gaining a deeper understanding of neural networks.

## B  USEFUL LEMMAS

**Lemma B.1.** *(Lemma 6 in Jin et al. (2022)) Suppose that $\{X_n\} \in \mathbb{R}^d$ is a non-negative sequence of random variables. If it holds that $\sum_{n=0}^{\infty} \mathbb{E}\left(X_n\right) < +\infty$, then $\sum_{n=0}^{\infty} X_n < +\infty$ holds almost surely.*

**Lemma B.2.** *(Wang et al., 2019) Suppose that $\{X_n\} \in \mathbb{R}^d$ is an $\mathcal{L}_2$ martingale difference sequence, and $(X_n, \mathcal{F}_n)$ is an adaptive process. Suppose it holds that*

$$\sum_{n=1}^{\infty} \mathbb{E}(\|X_n\|^2) < +\infty, \quad or \quad \sum_{n=1}^{\infty} \mathbb{E}\left(\|X_n\|^2 \big| \mathscr{F}_{n-1}\right) < +\infty.$$

*Then, it holds $\sum_{k=0}^{\infty} X_k < +\infty$ almost surely.*

**Lemma B.3.** *(Lemma 13 in Jin et al. (2022)) Consider the AdaGrad update (1) under our problem setting given in Section 2 and Assumption 3.1. It holds that for any $\alpha_0 > 0, \theta_1$,*

$$\sum_{k=3}^{n} \mathbb{E}\left(\frac{\left\|\nabla g(\theta_k)\right\|^2}{S_{k-1}^{\frac{1}{2}+\epsilon}}\right) < +\infty.$$

**Lemma B.4.** *Consider the AdaGrad update (1) under our problem setting in Section 2 and Assumption 3.1. We have for any $\alpha_0 > 0, \theta_1$, there is*

$$\sum_{n=2}^{+\infty} \mathbb{E}\left(\frac{\left\|\nabla g(\theta_n)\right\|^2}{\sqrt{S_{n-1}}}\right) < +\infty, \quad and \quad \sum_{n=2}^{+\infty} \frac{\left\|\nabla g(\theta_n)\right\|^2}{\sqrt{S_{n-1}}} < +\infty \ \ a.s..$$

**Lemma B.5.** *Consider the AdaGrad update (1) under our problem setting in Section 2 and Assumption 3.1, and function $f(\theta) := 1 - \frac{\theta^{\top}\hat{\theta}^*}{\|\theta\|+1}$, where $\hat{\theta}^*$ ($\|\hat{\theta}^*\| = 1$) is a max margin vector. We have for any $\alpha_0 > 0, \theta_1, \exists \, r_0 > 0, \, \tilde{M}_0 > 0$, such that*

$$\mathbb{E}\left(\frac{(\nabla f(\theta_n))^{\top}\nabla g(\theta_n)}{\sqrt{S_{n-1}}}\right)$$

$$\leq \mathbb{E}\left(\frac{f_{x_{i_n}}(\theta_n, x_{i_n})\|\theta_n\|}{N(\|\theta_n\|+1)^2 \sqrt{S_{n-1}}}\left(\frac{1}{2}\psi_{i_n}\left(\frac{\theta_n^{\top}x_{i_n}}{\|\theta_n\|} - \hat{\theta}^{*\top}x_{i_n}\right) + \frac{r_0}{\|\theta_n\|+1} + \frac{\tilde{M}_0}{e^{s'\|\theta_n\|}}\right)\right)$$

$$+ \frac{N\max_{1\leq i\leq N}\{\|x_i\|^2\}}{4c\ln 2} \cdot \mathbb{E}\left(\frac{\|\nabla g(\theta_n)\|^2}{\sqrt{S_{n-1}}}\right),$$

*where $\psi_i := sgn(y_i - 0.5)$ and*

$$f_{x_i}(\theta, x_i) := \begin{cases} f(\theta, x_i), & \text{if } y_i = 0, \\ 1 - f(\theta, x_i), & \text{if } y_i = 1, \end{cases}$$

$$f(\theta, x_i) := \frac{1}{1 + e^{-sgn(y_i - 0.5)\theta^{\top}x_i}}.$$

**Lemma B.6.** $x_1$ and $x_2$ are two d-dimensional. Then there is a vector $\theta \in \mathbb{R}^d$ which hold $|\theta^\top x_1|/\|\theta\| < |\theta^\top x_2|/\|\theta\|$. We assign $\theta^* := \arg\min_{\{\theta\,|\,|\theta^\top x_1|=|\theta^\top x_2|\}} \|\theta\|$. Then there exists $\delta_0 > 0$, $\hat{r} > 0$, such that

$$\left| \frac{\theta^\top x_2}{\|\theta\|} - \frac{\theta^* x_2}{\|\theta^*\|} \right| < \hat{r} \left| \frac{\theta^\top x_1}{\|\theta\|} - \frac{\theta^* x_1}{\|\theta^*\|} \right| \tag{7}$$

for any $\theta$ satisfying $\theta/\|\theta\| \in U(\theta^*, \delta_0)/\{\theta^*\}$.

**Lemma B.7.** (Lemma 10 in Jin et al. (2022)) Suppose $f(x) \in C^1$ $(x \in \mathbb{R}^N)$ with $f(x) > -\infty$ and its gradient satisfying the following Lipschitz condition

$$\left\| \nabla f(x) - \nabla f(y) \right\| \leq c\|x - y\|,$$

then $\forall\, x_0 \in \mathbb{R}^N$, there is

$$\left\| \nabla f(x_0) \right\|^2 \leq 2c\big(f(x_0) - f^*\big),$$

where $f^* = \inf_{x \in \mathbb{R}^N} f(x)$.

**Lemma B.8.** $\{\hat{x}_i, \hat{y}_i\}$ is a linear separable data set and $\hat{g}(\theta)$ is the loss of logistic regression. Then we have that if $\theta$ is a margin vertor of $\{\hat{x}_i, \hat{y}_i\}$, the loss function will hold that

$$k_1 \hat{g}(\theta) \leq \|\nabla \hat{g}(\theta)\| \leq k_2 \hat{g}(\theta),$$

where $k_1 > 0$, $k_2 > 2$ are two constant.

**Lemma B.9.** For a linear separable data set $S$, we assum its max-margin vertor as $\theta^*/\|\theta^*\|$, Then exists a constant $\tilde{\delta}_0 > 0$, making for any $\theta/\|\theta\| \in U(\theta^*/\|\theta^*\|, \tilde{\delta}_0)$, $\theta/\|\theta\|$ is a margin vector.

**Lemma B.10.** If a vector $\theta \in \mathbb{R}^d$ is not a margin vector, it will make $g(\theta) > (\ln 2)/N$.

## C  PROOFS OF LEMMAS AND THEOREMS

### C.1  THE PROOF OF LEMMA B.4

*Proof.* Based on calculations, it is easy to observe that when $\|\nabla g(\theta)\| \to 0$, there is $\|\nabla^2 g(\theta)\| = \Theta(\|\nabla g(\theta)\|)$ (Here, the norm represents the maximum eigenvalue of the Hessian matrix.). That means existing $\tilde{d}_0 > 0$, $\tilde{\delta}_1 > 0$, such that for any $\|\nabla g(\theta)\| < \tilde{\delta}_1$, there is $\|\nabla^2 g(\theta)\| \leq \tilde{d}_0\|\nabla g(\theta)\|$. Then we assign $\delta_1 := \min\{\ln 2/N, \tilde{\delta}_1/k_2\}$. Lemma B.8 and Lemma B.10, we know when $g(\theta) < \delta_1$, there is $\|\nabla^2 g(\theta)\| \leq \tilde{d}_0\|\nabla g(\theta)\|$. Then we define $S^{(\delta_2)} := \{\theta | g(\theta) < \delta_2 := \min\{\delta_0, \delta_1\}\}$, where $\delta_0$ defined in Assumption 1. We know that within the set $S^{(\delta_2)}$, the Hessian matrix is Lipschitz continuous. We define $\hat{c}$ as the Lipschitz constant of the Hessian matrix. We consider an event $\mathcal{B}_n := \{\theta_n \in S^{(\delta_2)}\}$. Meanwhile, we assign its complementary event as $\mathcal{B}_n^{(-)}$. Then, through the third-order Taylor expansion, we have

$$\mathbf{1}_{\mathcal{B}_n}\big(g(\theta_{n+1}) - g(\theta_n)\big) \leq -\mathbf{1}_{\mathcal{B}_n}\alpha_0 S_n^{-\frac{1}{2}}\nabla g(\theta_n)^\top \nabla g(\theta_n, \xi_n) + \mathbf{1}_{\mathcal{B}_n}\frac{\alpha_0^2\|\nabla^2 g(\theta_n)\| \cdot \|\nabla g(\theta_n, \xi_n)\|^2}{S_n}$$

$$+ \mathbf{1}_{\mathcal{B}_n}\frac{\hat{c}\alpha_0^3\|\nabla g(\theta_n, \xi_n)\|^3}{S_n^{\frac{3}{2}}}.$$

Combining Assumption 3.1, we can get

$$\mathbf{1}_{\mathcal{B}_n}\big(g(\theta_{n+1}) - g(\theta_n)\big) \leq -\mathbf{1}_{\mathcal{B}_n}\frac{\alpha_0 \nabla g(\theta_n)^\top \nabla g(\theta_n, \xi_n)}{\sqrt{S_{n-1}}} + \mathbf{1}_{\mathcal{B}_n}t_0\left(\frac{1}{\sqrt{S_{n-1}}} - \frac{1}{\sqrt{S_n}}\right)$$

$$+ \mathbf{1}_{\mathcal{B}_n}\frac{\tilde{d}_0\alpha_0^2\|\nabla g(\theta_n)\| \cdot \|\nabla g(\theta_n, \xi_n)\|^2}{S_n} + \mathbf{1}_{\mathcal{B}_n}\frac{\hat{c}\hat{K}\alpha_0^3\|\nabla g(\theta_n, \xi_n)\|^2}{S_n^{\frac{3}{2}}}, \tag{8}$$

where $t_0 = \alpha_0\hat{\delta}_1\hat{K}$. For the third term on the right side, we have

$$\mathbf{1}_{\mathcal{B}_n}\frac{\tilde{d}_0\alpha_0^2\|\nabla g(\theta_n)\| \cdot \|\nabla g(\theta_n, \xi_n)\|^2}{S_n} = \mathbf{1}\left(\|\nabla g(\theta_n)\| < \frac{2\alpha_0\tilde{d}_0\hat{K}_0}{\sqrt{S_n}}\right)\mathbf{1}_{\mathcal{B}_n}\frac{\tilde{d}_0\alpha_0^2\|\nabla g(\theta_n)\| \cdot \|\nabla g(\theta_n, \xi_n)\|^2}{S_n}$$

$$+ \mathbf{1}\left(\|\nabla g(\theta_n)\| \geq \frac{2\alpha_0\tilde{d}_0\hat{K}_0}{\sqrt{S_n}}\right)\mathbf{1}_{\mathcal{B}_n}\frac{\tilde{d}_0\alpha_0^2\|\nabla g(\theta_n)\| \cdot \|\nabla g(\theta_n, \xi_n)\|^2}{S_n}.$$

$$\tag{9}$$

Then we can acquire

$$\mathbf{1}_{\mathcal{B}_n} \frac{\tilde{d}_0 \alpha_0^2 \|\nabla g(\theta_n)\| \cdot \|\nabla g(\theta_n, \xi_n)\|^2}{S_n} \le 2\hat{d}_0^2 \alpha_0^3 \hat{K}^2 \mathbf{1}_{\mathcal{B}_n} \frac{\|\nabla g(\theta_n, \xi_n)\|^2}{S_n^{\frac{3}{2}}}$$
$$+ \frac{\alpha_0}{2\hat{K}^2} \mathbf{1}_{\mathcal{B}_n} \frac{\|\nabla g(\theta_n)\|^2 \|\nabla g(\theta_n, \xi_n)\|^2}{\sqrt{S_n}}.$$

Substitute above inequity into Equation (8), and make the mathematical expectation, getting

$$\mathbb{E}\left(\mathbf{1}_{\mathcal{B}_n}\big(g(\theta_{n+1}) - g(\theta_n)\big)\right) \le -\frac{1}{2}\mathbb{E}\left(\mathbf{1}_{\mathcal{B}_n} \frac{\alpha_0 \|\nabla g(\theta_n)\|^2}{\sqrt{S_{n-1}}}\right) + t_0 \mathbb{E}\left(\frac{1}{\sqrt{S_{n-1}}} - \frac{1}{\sqrt{S_n}}\right)$$
$$+ \mathbb{E}\left(\frac{(\hat{c}\hat{K} + 2\hat{d}_0^2\hat{K}^2)\|\nabla g(\theta_n, \xi_n)\|^2}{S_n^{\frac{3}{2}}}\right). \tag{10}$$

We make $\mathbb{E}\left(\mathbf{1}_{\mathcal{B}_n}\big(g(\theta_{n+1})\big)\right)$ to $\mathbb{E}\left(\mathbf{1}_{\mathcal{B}_{n+1}}\big(g(\theta_{n+1})\big)\right) + \mathbb{E}\left((\mathbf{1}_{\mathcal{B}_n} - \mathbf{1}_{\mathcal{B}_{n+1}})\big(g(\theta_{n+1})\big)\right)$, acquiring

$$\mathbb{E}\left(\mathbf{1}_{\mathcal{B}_{n+1}}\big(g(\theta_{n+1})\big)\right) - \mathbb{E}\left(\mathbf{1}_{\mathcal{B}_n} g(\theta_n)\right)$$
$$\le -\frac{1}{2}\mathbb{E}\left(\mathbf{1}_{\mathcal{B}_n} \frac{\alpha_0 \|\nabla g(\theta_n)\|^2}{\sqrt{S_{n-1}}}\right) + t_0 \mathbb{E}\left(\frac{1}{\sqrt{S_{n-1}}} - \frac{1}{\sqrt{S_n}}\right) \tag{11}$$
$$+ \mathbb{E}\left(\frac{(\hat{c}\hat{K} + 2\hat{d}_0^2\hat{K}^2)\|\nabla g(\theta_n, \xi_n)\|^2}{S_n^{\frac{3}{2}}}\right) - \mathbb{E}\left((\mathbf{1}_{\mathcal{B}_n} - \mathbf{1}_{\mathcal{B}_{n+1}})\big(g(\theta_{n+1})\big)\right).$$

We notice

$$-\mathbb{E}\left(\big(\mathbf{1}_{\mathcal{B}_n} - \mathbf{1}_{\mathcal{B}_{n+1}}\big)g(\theta_{n+1})\right)$$
$$= -\mathbb{E}\left(\big(\mathbf{1}_{\mathcal{B}_n} - \mathbf{1}_{\mathcal{B}_n}\mathbf{1}_{\mathcal{B}_{n+1}}\big)g(\theta_{n+1})\right) - \mathbb{E}\left(\big(\mathbf{1}_{\mathcal{B}_n}\mathbf{1}_{\mathcal{B}_{n+1}} - \mathbf{1}_{\mathcal{B}_{n+1}}\big)g(\theta_{n+1})\right)$$
$$\le \min\{\delta_0, \delta_1\} \cdot \mathbb{E}\left(\big(\mathbf{1}_{\mathcal{B}_n} - \mathbf{1}_{\mathcal{B}_n}\mathbf{1}_{\mathcal{B}_{n+1}}\big)\right) + \min\{\delta_0, \delta_1\} \cdot \mathbb{E}\left(\big(\mathbf{1}_{\mathcal{B}_n}\mathbf{1}_{\mathcal{B}_{n+1}} - \mathbf{1}_{\mathcal{B}_{n+1}}\big)\right)$$
$$= \min\{\delta_0, \delta_1\} \cdot \mathbb{E}\left(\mathbf{1}_{\mathcal{B}_n} - \mathbf{1}_{\mathcal{B}_{n+1}}\right).$$

we getting

$$\mathbb{E}\left(\mathbf{1}_{\mathcal{B}_{n+1}}\big(g(\theta_{n+1})\big)\right) - \mathbb{E}\left(\mathbf{1}_{\mathcal{B}_n} g(\theta_n)\right) \le -\frac{1}{2}\mathbb{E}\left(\mathbf{1}_{\mathcal{B}_n} \frac{\alpha_0 \|\nabla g(\theta_n)\|^2}{\sqrt{S_{n-1}}}\right) + t_0 \mathbb{E}\left(\frac{1}{\sqrt{S_{n-1}}} - \frac{1}{\sqrt{S_n}}\right)$$
$$+ \mathbb{E}\left(\frac{(\hat{c}\hat{K} + 2\hat{d}_0^2\hat{K}^2)\|\nabla g(\theta_n, \xi_n)\|^2}{S_n^{\frac{3}{2}}}\right) - \min\{\delta_0, \delta_1\} \cdot \mathbb{E}\left(\mathbf{1}_{\mathcal{B}_n} - \mathbf{1}_{\mathcal{B}_{n+1}}\right).$$

Then we make a sum, acquiring

$$\sum_{n=1}^{+\infty} \mathbb{E}\left(\mathbf{1}_{\mathcal{B}_n} \frac{\alpha_0 \|\nabla g(\theta_n)\|^2}{\sqrt{S_{n-1}}}\right) < +\infty. \tag{12}$$

Then we consider the case when $\mathcal{B}_n^{(-)}$ occurs. We know $\nabla g$ must hold the Lipschitz condition; we assign its Lipschitz constant as $c$. Then we get

$$\mathbf{1}_{\mathcal{B}_n^{(-)}}\big(g(\theta_{n+1}) - g(\theta_n)\big) \le -\mathbf{1}_{\mathcal{B}_n^{(-)}} \alpha_0 S_n^{-\frac{1}{2}} \nabla g(\theta_n)^\top \nabla g(\theta_n, \xi_n) + \mathbf{1}_{\mathcal{B}_n^{(-)}} \frac{\alpha_0^2 c \|\nabla g(\theta_n, \xi_n)\|^2}{S_n}.$$

First, we have

$$\mathbf{1}_{\mathcal{B}_n}^{(-)}\big(g(\theta_{n+1}) - g(\theta_n)\big) \le -\mathbf{1}_{\mathcal{B}_n}^{(-)} \frac{\alpha_0 \nabla g(\theta_n)^T \nabla g(\theta_n, \xi_n)}{\sqrt{S_n}} + \mathbf{1}_{\mathcal{B}_n}^{(-)} \frac{c\alpha_0^2}{2} \frac{\big\|\nabla g(\theta_n, \xi_n)\big\|^2}{S_n}$$

$$\le -\mathbf{1}_{\mathcal{B}_n}^{(-)} \frac{\alpha_0}{2}\left(\frac{1}{M+1}\frac{\big\|\nabla g(\theta_n, \xi_n)\big\|^2}{\sqrt{S_n}} + (M+1)\frac{\big\|\nabla g(\theta_n)\big\|^2}{\sqrt{S_n}}\right)$$

$$+ \mathbf{1}_{\mathcal{B}_n}^{(-)} \frac{\alpha_0}{2} \frac{1}{\sqrt{S_{n-1}}}\left\|\frac{1}{\sqrt{M+1}}\nabla g(\theta_n, \xi_n) - \sqrt{M+1}\nabla g(\theta_n)\right\|^2 + \mathbf{1}_{\mathcal{B}_n}^{(-)} \frac{c\alpha_0^2}{2} \frac{\big\|\nabla g(\theta_n, \xi_n)\big\|^2}{S_n}$$

$$\le \mathbf{1}_{\mathcal{B}_n}^{(-)} \frac{\alpha_0}{2}(M+1)\left(\frac{\big\|\nabla g(\theta_{n-1})\big\|^2}{\sqrt{S_{n-1}}} - \frac{\big\|\nabla g(\theta_n)\big\|^2}{\sqrt{S_n}}\right)$$

$$+ \mathbf{1}_{\mathcal{B}_n}^{(-)} \frac{\alpha_0}{2}\left(\frac{1}{M+1}\frac{\big\|\nabla g(\theta_n, \xi_n)\big\|^2}{\sqrt{S_{n-1}}} + \mathbf{1}_{\mathcal{B}_n}^{(-)} \frac{(M-1)\big\|\nabla g(\theta_n)\big\|^2}{\sqrt{S_{n-1}}} - \frac{(M+1)\big\|\nabla g(\theta_{n-1})\big\|^2}{\sqrt{S_{n-1}}}\right)$$

$$+ \mathbf{1}_{\mathcal{B}_n}^{(-)} \frac{c\alpha_0^2}{2} \frac{\big\|\nabla g(\theta_n, \xi_n)\big\|^2}{S_n} + X_n, \tag{13}$$

where $X_n$ is defined as follow

$$X_n := \mathbf{1}_{\mathcal{B}_n}^{(-)} \frac{\alpha_0}{\sqrt{S_{n-1}}}\nabla g(\theta_n)^T\big(\nabla g(\theta_n) - \nabla g(\theta_n, \xi_n)\big),$$

and $M := 2M_0 + 2(a/k_1^2\delta_2^2) - 1$. Then we can find

$$\big\|\nabla g(\theta_n)\big\|^2 = \big\|\nabla g(\theta_{n-1}) + \big(\nabla g(\theta_n) - \nabla g(\theta_{n-1})\big)\big\|^2$$

$$\le \big\|\nabla g(\theta_{n-1})\big\|^2 + \frac{2\alpha_0 c}{\sqrt{S_{n-1}}}\big\|\nabla g(\theta_{n-1})\big\|\big\|\nabla g(\theta_{n-1}, \xi_{n-1})\big\| + c^2\alpha_0^2 \frac{\big\|\nabla g(\theta_{n-1}, \xi_{n-1})\big\|^2}{S_{n-1}}. \tag{14}$$

Then we multiple $M + \frac{1}{2}$ on the both side of above inequity, acquiring

$$\left(M + \frac{1}{2}\right)\big\|\nabla g(\theta_n)\big\|^2 \le \left(M + \frac{1}{2}\right)\big\|\nabla g(\theta_{n-1})\big\|^2 + \left(M + \frac{1}{2}\right)\frac{2\alpha_0 c}{\sqrt{S_{n-1}}}\big\|\nabla g(\theta_{n-1})\big\|\big\|\nabla g(\theta_{n-1}, \xi_{n-1})\big\|$$

$$+ \left(M + \frac{1}{2}\right)c^2\alpha_0^2\frac{\big\|\nabla g(\theta_{n-1}, \xi_{n-1})\big\|^2}{S_{n-1}}.$$

Noting

$$\left(M + \frac{1}{2}\right)\frac{2\alpha_0 c}{\sqrt{S_{n-1}}}\big\|\nabla g(\theta_{n-1})\big\|\big\|\nabla g(\theta_{n-1}, \xi_{n-1})\big\| \le \frac{1}{2}\big\|\nabla g(\theta_{n-1})\big\|^2 + 2\left(M + \frac{1}{2}\right)^2\alpha_0^2 c^2 \frac{\big\|\nabla g(\theta_{n-1}, \xi_{n-1})\big\|^2}{S_{n-1}}.$$

We get that

$$\left(M + \frac{1}{2}\right)\big\|\nabla g(\theta_n)\big\|^2 \le (M+1)\big\|\nabla g(\theta_{n-1})\big\|^2 + \left(2\left(M + \frac{1}{2}\right)^2\alpha_0^2 c^2 + \left(M + \frac{1}{2}\right)c^2\alpha_0^2\right)\frac{\big\|\nabla g(\theta_{n-1}, \xi_{n-1})\big\|^2}{S_{n-1}},$$

that is

$$(M-1)\big\|\nabla g(\theta_n)\big\|^2 + \frac{M_0 + \frac{a}{k_1^2\delta_2^2}}{M+1}\big\|\nabla g(\theta_n)\big\|^2$$

$$\le -\big\|\nabla g(\theta_n)\big\|^2 + (M+1)\big\|\nabla g(\theta_{n-1})\big\|^2 + \left(2\left(M + \frac{1}{2}\right)^2\alpha_0^2 c^2 + \left(M + \frac{1}{2}\right)c^2\alpha_0^2\right)\frac{\big\|\nabla g(\theta_{n-1}, \xi_{n-1})\big\|^2}{S_{n-1}}.$$

Then we multiple $\mathbf{1}_{\mathcal{B}_n}^{(-)}/\sqrt{S_{n-1}}$ on both side of above inequity, and noting where $\mathbf{1}_{\mathcal{B}_n}^{(-)} = 1$, there is

$$\frac{M_0 + \frac{a}{k_1^2\delta_2^2}}{M+1}\big\|\nabla g(\theta_n)\big\|^2 \ge \frac{1}{M+1}\,\mathbb{E}(\|\nabla g(\theta_n, \xi_n)\|^2|\mathscr{F}_n),$$

getting

$$
\begin{aligned}
(M-1)\,\mathbb{E}&\left(\mathbf{1}_{\mathcal{B}_n}^{(-)}\frac{\left\|\nabla g(\theta_n)\right\|^2}{\sqrt{S_{n-1}}}\right) + \frac{1}{M+1}\,\mathbb{E}\left(\mathbf{1}_{\mathcal{B}_n}^{(-)}\frac{\left\|\nabla g(\theta_n,\xi_n)\right\|^2}{S_{n-1}}\right) \\
&\leq -\,\mathbb{E}\left(\mathbf{1}_{\mathcal{B}_n}^{(-)}\frac{\left\|\nabla g(\theta_n)\right\|^2}{\sqrt{S_{n-1}}}\right) + (M+1)\,\mathbb{E}\left(\mathbf{1}_{\mathcal{B}_n}^{(-)}\frac{\left\|\nabla g(\theta_{n-1})\right\|^2}{\sqrt{S_{n-1}}}\right) \\
&\quad + \left(2\left(M+\tfrac{1}{2}\right)^2\alpha_0^2 c^2 + \left(M+\tfrac{1}{2}\right)c^2\alpha_0^2\right)\mathbb{E}\left(\mathbf{1}_{\mathcal{B}_n}^{(-)}\frac{\left\|\nabla g(\theta_{n-1},\xi_{n-1})\right\|^2}{S_{n-1}^{\frac{3}{2}}}\right).
\end{aligned}
\tag{15}
$$

Substitute it into Equation (13), we get

$$
\begin{aligned}
\mathbb{E}\left(\mathbf{1}_{\mathcal{B}_{n+1}}^{(-)}\left(g(\theta_{n+1})\right) - \mathbb{E}\left(\mathbf{1}_{\mathcal{B}_n}^{(-)}g(\theta_n)\right)\right) &\leq -\frac{\alpha_0}{2}\,\mathbb{E}\left(\mathbf{1}_{\mathcal{B}_n}^{(-)}\frac{\left\|\nabla g(\theta_n)\right\|^2}{\sqrt{S_{n-1}}}\right) \\
&+ \left(2\left(M+\tfrac{1}{2}\right)^2\alpha_0^2 c^2 + \left(M+\tfrac{1}{2}\right)c^2\alpha_0^2\right)\mathbb{E}\left(\mathbf{1}_{\mathcal{B}_n}^{(-)}\frac{\left\|\nabla g(\theta_{n-1},\xi_{n-1})\right\|^2}{S_{n-1}^{\frac{3}{2}}}\right) \\
&+ \mathbb{E}\left(\mathbf{1}_{\mathcal{B}_n}^{(-)}\frac{\alpha_0}{2}(M+1)\left(\frac{\left\|\nabla g(\theta_{n-1})\right\|^2}{\sqrt{S_{n-1}}} - \frac{\left\|\nabla g(\theta_n)\right\|^2}{\sqrt{S_n}}\right)\right) + \frac{c\alpha_0^2}{2}\,\mathbb{E}\left(\frac{\left\|\nabla g(\theta_n,\xi_n)\right\|^2}{S_n}\right) \\
&+ \mathbb{E}\left((\mathbf{1}_{\mathcal{B}_n}-\mathbf{1}_{\mathcal{B}_{n+1}})\left(g(\theta_{n+1})\right)\right).
\end{aligned}
\tag{16}
$$

Then we use inequality $2a^T b \leq \lambda\|a\|^2 + \frac{1}{\lambda}\|b\|^2\ \ (\lambda>0)$ on Equation (14) to get

$$
\begin{aligned}
\left\|\nabla g(\theta_n)\right\|^2 - \left\|\nabla g(\theta_{n-1})\right\|^2 &\leq \frac{\left\|\nabla g(\theta_{n-1})\right\|^2}{2(M+1)} \\
&+ \frac{2\alpha_0^2 c^2(M+1)}{S_{n-1}}\left\|\nabla g(\theta_{n-1},\xi_{n-1})\right\|^2 + \frac{\alpha_0^2 c^2}{S_{n-1}}\left\|\nabla g(\theta_{n-1},\xi_{n-1})\right\|^2.
\end{aligned}
\tag{17}
$$

Multiple both sides of Equation (17) by $\mathbf{1}_{\mathcal{B}_n}^{(-)}/\sqrt{S_{n-1}}$ and notice $S_{n-2}\leq S_{n-1}\leq S_n$, then we have

$$
\begin{aligned}
\mathbf{1}_{\mathcal{B}_n}^{(-)}&\left(\frac{\left\|\nabla g(\theta_n)\right\|^2}{\sqrt{S_n}} - \frac{\left\|\nabla g(\theta_{n-1})\right\|^2}{\sqrt{S_{n-1}}}\right) \\
&\leq \frac{\mathbf{1}_{\mathcal{B}_n}^{(-)}\left\|\nabla g(\theta_{n-1})\right\|^2}{2(M+1)\sqrt{S_{n-2}}} + \frac{\alpha_0^2 c^2(2M+3)\mathbf{1}_{\mathcal{B}_n}^{(-)}}{S_{n-1}^{\frac{3}{2}}}\left\|\nabla g(\theta_{n-1},\xi_{n-1})\right\|^2 \\
&\leq \frac{\mathbf{1}_{\mathcal{B}_n}^{(-)}\mathbf{1}_{\mathcal{B}_{n-1}}^{(-)}\left\|\nabla g(\theta_{n-1})\right\|^2}{2(M+1)\sqrt{S_{n-2}}} + (\mathbf{1}_{\mathcal{B}_n}^{(-)} - \mathbf{1}_{\mathcal{B}_{n-1}}^{(-)}\mathbf{1}_{\mathcal{B}_n}^{(-)})\frac{\left\|\nabla g(\theta_{n-1})\right\|^2}{2(M+1)\sqrt{S_{n-2}}} \\
&\quad + \frac{\alpha_0^2 c^2(2M+3)\mathbf{1}_{\mathcal{B}_n}^{(-)}}{S_{n-1}^{\frac{3}{2}}}\left\|\nabla g(\theta_{n-1},\xi_{n-1})\right\|^2 \\
&\leq \frac{\mathbf{1}_{\mathcal{B}_{n-1}}^{(-)}\left\|\nabla g(\theta_{n-1})\right\|^2}{2(M+1)\sqrt{S_{n-2}}} + \mathbf{1}_{\mathcal{B}_{n-1}}\frac{\left\|\nabla g(\theta_{n-1})\right\|^2}{2(M+1)\sqrt{S_{n-2}}} \\
&\quad + \frac{\alpha_0^2 c^2(2M+3)\mathbf{1}_{\mathcal{B}_n}^{(-)}}{S_{n-1}^{\frac{3}{2}}}\left\|\nabla g(\theta_{n-1},\xi_{n-1})\right\|^2.
\end{aligned}
\tag{18}
$$

Substitute Equation (18) into Equation (15), acquiring

$$
\begin{aligned}
\mathbb{E}\left(\mathbf{1}_{\mathcal{B}_{n+1}}^{(-)}\big(g(\theta_{n+1})\big)\right) - \mathbb{E}\left(\mathbf{1}_{\mathcal{B}_n}^{(-)}g(\theta_n)\big)\right) &\leq -\frac{\alpha_0}{4}\,\mathbb{E}\left(\mathbf{1}_{\mathcal{B}_n}^{(-)}\frac{\left\|\nabla g(\theta_n)\right\|^2}{\sqrt{S_{n-1}}}\right) \\
&+ \left(2\left(M+\frac{1}{2}\right)^2\alpha_0^2 c^2 + \left(M+\frac{1}{2}\right)c^2\alpha_0^2\right)\mathbb{E}\left(\mathbf{1}_{\mathcal{B}_n}^{(-)}\frac{\left\|\nabla g(\theta_{n-1},\xi_{n-1})\right\|^2}{S_{n-1}^{\frac{3}{2}}}\right) \\
&+ \frac{c\alpha_0^2}{2}\,\mathbb{E}\left(\frac{\left\|\nabla g(\theta_n,\xi_n)\right\|^2}{S_n}\right) + \mathbb{E}\left(\frac{\alpha_0^2 c^2 (2M+3)(M+1)\mathbf{1}_{\mathcal{B}_n}^{(-)}}{2 S_{n-1}^{\frac{3}{2}}}\left\|\nabla g(\theta_{n-1},\xi_{n-1})\right\|^2\right) \\
&+ \mathbb{E}\left((\mathbf{1}_{\mathcal{B}_n}-\mathbf{1}_{\mathcal{B}_{n+1}})\big(g(\theta_{n+1})\big)\right) + \mathbb{E}\left(\mathbf{1}_{\mathcal{B}_{n-1}}\frac{\left\|\nabla g(\theta_{n-1})\right\|^2}{2(M+1)\sqrt{S_{n-2}}}\right).
\end{aligned}
\tag{19}
$$

We know when $\mathcal{B}_n^{(-)}$ occurs, through Lemma B.8, there is $\|\nabla g(\theta)\| > \delta_2' := k_1 \cdot \min\{\delta_0,\delta_1\}$. That means

$$
\mathbb{E}\left(\|\nabla g(\theta_n,\xi_n)\|^2\big|\mathscr{F}_n\right) \leq M\|\nabla g(\theta_n)\|^2 + a \leq \left(M+\frac{a}{\delta_2'^2}\right)\|\nabla g(\theta_n)\|^2.
$$

so we get

$$
\sum_{k=1}^{n}\mathbb{E}\left(\mathbf{1}_{B_n^{(-)'}}\frac{\left\|\nabla g(\theta_k,\xi_k)\right\|^2}{S_k}\right) \leq \left(M+\frac{a}{\delta_2^2}\right)\mathbb{E}\left(\frac{\left\|\nabla g(\theta_k)\right\|^2}{S_k}\right).
$$

Through Lemma B.3, we can get

$$
\mathbb{E}\left(\frac{\left\|\nabla g(\theta_k)\right\|^2}{S_k}\right) < \tilde{o}\,\mathbb{E}\left(\frac{\left\|\nabla g(\theta_k)\right\|^2}{S_k^{\frac{3}{4}}}\right) < +\infty.
$$

We back to Equation (11), we can get

$$
\begin{aligned}
\mathbb{E}\left((\mathbf{1}_{\mathcal{B}_n}-\mathbf{1}_{\mathcal{B}_{n+1}})\big(g(\theta_{n+1})\big)\right) &\leq \mathbb{E}\left(\mathbf{1}_{\mathcal{B}_n}g(\theta_n)\big)\right) - \mathbb{E}\left(\mathbf{1}_{\mathcal{B}_{n+1}}\big(g(\theta_{n+1})\big) + t_0\,\mathbb{E}\left(\frac{1}{\sqrt{S_{n-1}}}-\frac{1}{\sqrt{S_n}}\right)\right) \\
&+ \mathbb{E}\left(\frac{(\hat{c}\hat{K}+2\hat{d}_0^2\hat{K}^2)\|\nabla g(\theta_n,\xi_n)\|^2}{S_n^{\frac{3}{2}}}\right),
\end{aligned}
\tag{20}
$$

which means

$$
\sum_{n=1}^{+\infty}\mathbb{E}\left((\mathbf{1}_{\mathcal{B}_n}-\mathbf{1}_{\mathcal{B}_{n+1}})\big(g(\theta_{n+1})\big)\right) < +\infty
\tag{21}
$$

Substitute Equation (20) and Equation (21) into Equation (19), and make a sum, we get

$$
\sum_{n=1}^{+\infty}\mathbb{E}\left(\mathbf{1}_{\mathcal{B}_n}^{(-)}\frac{\alpha_0\|\nabla g(\theta_n)\|^2}{\sqrt{S_{n-1}}}\right) < +\infty.
\tag{22}
$$

Combine Equation (12) and Equation (22), we get

$$
\sum_{n=1}^{+\infty}\mathbb{E}\left(\frac{\alpha_0\|\nabla g(\theta_n)\|^2}{\sqrt{S_{n-1}}}\right) < +\infty.
$$

With this, we complete the result. $\qquad\square$

## C.2 THE PROOF OF LEMMA B.8

*Proof.* We can get

$$
\hat{g}(\theta) = -\frac{1}{N}\sum_{i=1}^{N}\left(\hat{y}_i\ln\left(\frac{1}{1+e^{\mathrm{sgn}(\hat{y}_i-0.5)\theta^\top\hat{x}_i}}\right) + (1-\hat{y}_i)\ln\left(1-\frac{1}{1+e^{\mathrm{sgn}(\hat{y}_i-0.5)\theta^\top\hat{x}_i}}\right)\right).
$$

Due to $\theta$ is a margin vector, we can get

$$\hat{g}(\theta) = -\frac{1}{N} \sum_{i=1}^{N} \ln\left(1 - \frac{1}{1 + e^{|\theta^\top \hat{x}_i|}}\right).$$

Since $1/(1 + e^{|\theta^\top \hat{x}_i|}) \in (0, 1/2)$, we can get following inequality

$$-\frac{2\ln 2}{1 + e^{|\theta^\top \hat{x}_i|}} \leq \ln\left(1 - \frac{1}{1 + e^{|\theta^\top \hat{x}_i|}}\right) \leq -\frac{1}{1 + e^{|\theta^\top \hat{x}_i|}}.$$

That means

$$\frac{1}{N} \sum_{i=1}^{N} \frac{1}{1 + e^{|\theta^\top \hat{x}_i|}} \leq \hat{g}(\theta) \leq \frac{2\ln 2}{N} \sum_{i=1}^{N} \frac{1}{1 + e^{|\theta^\top \hat{x}_i|}}. \tag{23}$$

On the other hand, we can calculate

$$\|\nabla \hat{g}(\theta)\| = \frac{1}{N} \left\| \sum_{i=1}^{N} \left(\frac{1}{1 + e^{-\theta^\top \hat{x}_i}} - y_i\right) \hat{x}_i \right\|.$$

Due to $\theta$ is a margin vector, we can get

$$\|\nabla \hat{g}(\theta)\| = \frac{1}{N} \left\| \sum_{i=1}^{N} \frac{-\operatorname{sgn}(y_i - 0.5)}{1 + e^{|\theta^\top \hat{x}_i|}} \hat{x}_i \right\|.$$

First, we use the norm inequality, getting

$$\|\nabla \hat{g}(\theta)\| = \frac{1}{N} \left\| \sum_{i=1}^{N} \frac{-\operatorname{sgn}(y_i - 0.5)}{1 + e^{|\theta^\top \hat{x}_i|}} \hat{x}_i \right\| \leq \frac{\max_{1 \leq i \leq N} \{\|\hat{x}_i\|\}}{N} \cdot \sum_{i=1}^{N} \frac{1}{1 + e^{|\theta^\top \hat{x}_i|}}.$$

Second, we assume $\theta^*/\|\theta^*\|$ is the max margin vector of this data set, we getting

$$\|\nabla g(\theta)\| = \frac{1}{N} \left\| \sum_{i=1}^{N} \frac{-\operatorname{sgn}(y_i - 0.5)}{1 + e^{|\theta^\top \hat{x}_i|}} \hat{x}_i \right\| \geq \frac{1}{N} \left( \sum_{i=1}^{N} \frac{1}{1 + e^{|\theta^\top \hat{x}_i|}} \left| \frac{\theta^{*\top} \hat{x}_i}{\|\theta^*\|} \right| \right)$$

$$\geq \frac{d^*}{N} \sum_{i=1}^{N} \frac{1}{1 + e^{|\theta^\top \hat{x}_i|}}.$$

Then we can get

$$\frac{d^*}{N} \sum_{i=1}^{N} \frac{1}{1 + e^{|\theta^\top \hat{x}_i|}} \leq \|\nabla g(\theta)\| \leq \frac{\max_{1 \leq i \leq N} \{\|\hat{x}_i\|\}}{N} \cdot \sum_{i=1}^{N} \frac{1}{1 + e^{|\theta^\top \hat{x}_i|}}. \tag{24}$$

Combining Equation (23) and Equation (24), we can get the result.

### C.3 THE PROOF OF LEMMA B.10

Due to $\theta$ is not a margin vector of the data set $\{x_i, y_i\}_{i=1}^{N}$, we know that there is at least one data $(x_j, y_j)$ has a wrong classification which is formed by $\theta$. That means

$$-y_j \ln(\hat{y}_j) - (1 - y_j) \ln(1 - \hat{y}_j) > \ln 2,$$

so we can get

$$g(\theta) = -\frac{1}{N} \sum_{i=1}^{N} \left(y_i \ln(\hat{y}_i) + (1 - y_i) \ln(1 - \hat{y}_i)\right) > \frac{\ln 2}{N}.$$

$\square$

## C.4 THE PROOF OF LEMMA B.6

*Proof.* Obviously, since $\theta^* := \arg\min_{\{\theta | |\theta^\top x_1| = |\theta^\top x_2|\}} \|\theta\|$, we can get $\text{rank}\{x_1, x_2\} = \text{rank}\{x_1, x_2, \theta^*\}$. Then we assign $S := \text{span}\{x_1, x_2\}$. For any vector $\theta$, we assign the vector which $\theta$ projects on $S$ as $\theta'$. Without loss of generality, we can think $\theta^{*\top} x_1 = \theta^{*\top} x_2 > 0$. (if $\theta^{*\top} x_i < 0$, we can construct a new vector $x_i' := -x_i$ to substitute $x_i$.) Then we assign

$$\varphi := \arccos \frac{\theta'^\top \theta^*}{\|\theta'\|\|\theta^*\|},$$

$$\varphi_1 := \arccos \frac{\theta^{*\top} x_1}{\|\theta^*\|\|x_1\|}, \quad \varphi_2 := \arccos \frac{\theta^{*\top} x_2}{\|\theta^*\|\|x_2\|},$$

$$\phi := \arccos \frac{\theta^\top \theta'}{\|\theta\|\|\theta'\|},$$

In order to prove Equation (7), we just need to prove exists $\delta_0' > 0$, making the binary function

$$D(\varphi, \phi) = \frac{\|x_1\|}{\|x_2\|} \cdot \frac{\left| \cos(\varphi_1 - \varphi)\cos(\phi) - \cos(\varphi_1) \right|}{\cos(\varphi_2) - \cos(\varphi_2 + \varphi)\cos(\phi)} < \hat{r}, \ (\forall\, 0 < \varphi, \ \phi < \delta_0'). \tag{25}$$

Absolutely, when $\varphi$ and $\phi$ are small enough, we can cancel the absolute value, i.e.,

$$D(\varphi, \phi) = \frac{\|x_1\|}{\|x_2\|} \cdot \frac{\left| \cos(\varphi_1 - \varphi)\cos(\phi) - \cos(\varphi_1) \right|}{\cos(\varphi_2) - \cos(\varphi_2 + \varphi)\cos(\phi)}.$$

That means

$$\limsup_{\varphi \to 0, \phi \to 0} D(\varphi, \phi)$$

$$\leq \limsup_{\varphi \to 0, \phi \to 0} \frac{\|x_1\|}{\|x_2\|} \cdot \frac{\left| \cos(\varphi_1 - \varphi)\cos(\phi) - \cos(\varphi_1)\cos(\phi) \right| + \left| \cos(\varphi_1)\cos(\phi) - \cos(\varphi_1) \right|}{\cos(\varphi_2) - \cos(\varphi_2 + \varphi)\cos(\phi)}$$

$$\leq \frac{\|x_1\|}{\|x_2\|} \cdot \max\left\{ \frac{\sin(\varphi_1)}{\sin(\varphi_2)}, \frac{\cos(\varphi_1)}{\cos(\varphi_2)} \right\}.$$

That means we can take

$$\hat{r} := 2\frac{\|x_1\|}{\|x_2\|} \cdot \max\left\{ \frac{\sin(\varphi_1)}{\sin(\varphi_2)}, \frac{\cos(\varphi_1)}{\cos(\varphi_2)} \right\}$$

to make Equation (7) holding. $\qquad\square$

## C.5 THE PROOF OF LEMMA 4.1

*Proof.* The proof is similar to those to obtain the arguments in the proof of Lemma B.6. $\qquad\square$

## C.6 PROOF OF LEMMA 4.2

*Proof.* Given two unary function $y_1(x) = -1/\alpha |\ln x|^\alpha$ $(0 < x < 1/4)$, $y_2(x) = 1$ $(x > 1/2)$. We know that there is a smooth connecting function $y_3(x)$ $(1/4 \leq x \leq 1/2)$, making the following function

$$y(x) = \begin{cases} -1/\alpha |\ln x|^\alpha, & \text{if } x < \frac{\ln 2}{N} \\ 1, & \text{if } x > 1 \\ y_3(x), & \text{if } \frac{\ln 2}{N} \leq x \leq 1 \end{cases}$$

is an infinite order continuous function.

We construct a function

$$h(\theta) := y(g(\theta)), \tag{26}$$

and a set $S^{(\hat{\delta})} := \{\theta | 0 < g(\theta) < \hat{\delta}\}$. We make $\hat{\delta} = (\ln 2)/N$. Then we use the *taylor expansion* and the structure of $g$, getting that for any $\theta^{(1)} \in S^{(\hat{\delta})}$ and $\theta^{(2)} \in \mathbb{R}^d$, there exists three positive constants $d_0$, $d_1$ and $d_2$, making

$$h(\theta^{(2)}) - h(\theta^{(1)}) \leq \nabla h(\theta^{(1)})^\top (\theta^{(2)} - \theta^{(1)}) + \frac{d_0}{\|\theta^{(1)}\|^2} \|\theta^{(2)} - \theta^{(1)}\|^2$$

$$+ c_0 \|\theta^{(2)} - \theta^{(1)}\|^3, \tag{27}$$

where $\hat{c}$ is a constant that can not affect the result. For convenience, we assign

$$T_n := \frac{d_0 \alpha_0^2}{\|\theta_n\|^{1+\alpha}}.$$

We construct an event $\mathcal{A}_n := \{\theta_n \in S^{(\hat{\delta})}\}$ ($\hat{\delta} = (\ln 2)/N$). Combining Equation (27), we get

$$\mathbf{1}_{\mathcal{A}_n}\big(h(\theta_{n+1}) - h(\theta_n)\big) \leq \mathbf{1}_{\mathcal{A}_n} \nabla h(\theta_n)^\top (\theta_{n+1} - \theta_n) + \mathbf{1}_{\mathcal{A}_n} T_n \|\theta_{n+1} - \theta_n\|^2$$
$$+ \hat{c}\|\theta_{n+1} - \theta_n\|^3 \tag{28}$$
$$= -\mathbf{1}_{\mathcal{A}_n} \frac{\big(\nabla g(\theta_n)\big)^\top \nabla g(\theta_n, \xi_n)}{\sqrt{S_n} g(\theta_n)|\ln(g(\theta_n))|^{1+\alpha}} + \mathbf{1}_{\mathcal{A}_n} \frac{T_n \|\nabla g(\theta_n, \xi_n)\|^2}{S_n} + \mathbf{1}_{\mathcal{A}_n} \frac{c_0 \alpha_0^3 \|\nabla g(\theta_n, \xi_n)\|^3}{S_n^{\frac{3}{2}}}.$$

Then we get

$$\mathbf{1}_{\mathcal{A}_n}\big(h(\theta_{n+1}) - h(\theta_n)\big)$$
$$\leq -\mathbf{1}_{\mathcal{A}_n} \frac{\big(\nabla g(\theta_n)\big)^\top \nabla g(\theta_n, \xi_n)}{\sqrt{S_n} g(\theta_n)|\ln(g(\theta_n))|^{1+\alpha}} + \mathbf{1}_{\mathcal{A}_n} \frac{T_n \|\nabla g(\theta_n, \xi_n)\|^2}{S_n} + \mathbf{1}_{\mathcal{A}_n} \frac{c_0 \alpha_0^3 \|\nabla g(\theta_n, \xi_n)\|^3}{S_n^{\frac{3}{2}}}$$
$$= -\mathbf{1}_{\mathcal{A}_n} \frac{\big(\nabla g(\theta_n)\big)^\top \nabla g(\theta_n, \xi_n)}{\sqrt{S_{n-1}} g(\theta_n)|\ln(g(\theta_n))|^{1+\alpha}} + \mathbf{1}_{\mathcal{A}_n} \frac{\big(\nabla g(\theta_n)\big)^\top \nabla g(\theta_n, \xi_n)}{g(\theta_n)|\ln(g(\theta_n))|^{1+\alpha}} \left( \frac{1}{\sqrt{S_{n-1}}} - \frac{1}{\sqrt{S_n}} \right)$$
$$+ \mathbf{1}_{\mathcal{A}_n} \frac{T_n \|\nabla g(\theta_n, \xi_n)\|^2}{S_n} + \mathbf{1}_{\mathcal{A}_n} \frac{c_0 \alpha_0^3 \|\nabla g(\theta_n, \xi_n)\|^3}{S_n^{\frac{3}{2}}}.$$

Then we use an identical equation, i.e.,

$$\mathbf{1}_{\mathcal{A}_n} h(\theta_{n+1}) = \mathbf{1}_{\mathcal{A}_{n+1}} h(\theta_{n+1}) + \big(\mathbf{1}_{\mathcal{A}_n} - \mathbf{1}_{\mathcal{A}_{n+1}}\big) h(\theta_{n+1}),$$

getting

$$\mathbf{1}_{\mathcal{A}_{n+1}} h(\theta_{n+1}) - \mathbf{1}_{\mathcal{A}_n} h(\theta_n)$$
$$\leq -\mathbf{1}_{\mathcal{A}_n} \frac{\big(\nabla g(\theta_n)\big)^\top \nabla g(\theta_n, \xi_n)}{\sqrt{S_{n-1}} g(\theta_n)|\ln(g(\theta_n))|^{1+\alpha}} + \mathbf{1}_{\mathcal{A}_n} \frac{\big(\nabla g(\theta_n)\big)^\top \nabla g(\theta_n, \xi_n)}{g(\theta_n)|\ln(g(\theta_n))|^{1+\alpha}} \left( \frac{1}{\sqrt{S_{n-1}}} - \frac{1}{\sqrt{S_n}} \right) \tag{29}$$
$$+ \mathbf{1}_{\mathcal{A}_n} \frac{T_n \|\nabla g(\theta_n, \xi_n)\|^2}{S_n} + \mathbf{1}_{\mathcal{A}_n} \frac{c_0 \alpha_0^3 \|\nabla g(\theta_n, \xi_n)\|^3}{S_n^{\frac{3}{2}}} - \big(\mathbf{1}_{\mathcal{A}_n} - \mathbf{1}_{\mathcal{A}_{n+1}}\big) h(\theta_{n+1}).$$

We make the mathematical expectation on the both side of Equation (29), getting

$$\mathbb{E}\big(\mathbf{1}_{\mathcal{A}_{n+1}} h(\theta_{n+1})\big) - \mathbb{E}\big(\mathbf{1}_{\mathcal{A}_n} h(\theta_n)\big)$$
$$\leq -\mathbb{E}\left( \mathbf{1}_{\mathcal{A}_n} \frac{\big\|\nabla g(\theta_n)\big\|^2}{\sqrt{S_{n-1}} g(\theta_n)|\ln(g(\theta_n))|^{1+\alpha}} \right) + \mathbb{E}\left( \mathbf{1}_{\mathcal{A}_n} \frac{\big(\nabla g(\theta_n)\big)^\top \nabla g(\theta_n, \xi_n)}{g(\theta_n)|\ln(g(\theta_n))|^{1+\alpha}} \left( \frac{1}{\sqrt{S_{n-1}}} - \frac{1}{\sqrt{S_n}} \right) \right)$$
$$+ \mathbb{E}\left( \mathbf{1}_{\mathcal{A}_n} \frac{T_n \|\nabla g(\theta_n, \xi_n)\|^2}{S_n} \right) + \mathbb{E}\left( \mathbf{1}_{\mathcal{A}_n} \frac{c_0 \alpha_0^3 \|\nabla g(\theta_n, \xi_n)\|^3}{S_n^{\frac{3}{2}}} \right) - \mathbb{E}\left( \big(\mathbf{1}_{\mathcal{A}_n} - \mathbf{1}_{\mathcal{A}_{n+1}}\big) h(\theta_{n+1}) \right). \tag{30}$$

For the second item in Equation (30) right, through Assumption 3.1, there is

$$\mathbb{E}\left( \mathbf{1}_{\mathcal{A}_n} \frac{\big(\nabla g(\theta_n)\big)^\top \nabla g(\theta_n, \xi_n)}{g(\theta_n)|\ln(g(\theta_n))|^{1+\alpha}} \left( \frac{1}{\sqrt{S_{n-1}}} - \frac{1}{\sqrt{S_n}} \right) \right) \leq \tilde{\delta}_0 \mathbb{E}\left( \frac{1}{\sqrt{S_{n-1}}} - \frac{1}{\sqrt{S_n}} \right). \tag{31}$$

Next we get

$$-\mathbb{E}\left( \big(\mathbf{1}_{\mathcal{A}_n} - \mathbf{1}_{\mathcal{A}_{n+1}}\big) h(\theta_{n+1}) \right)$$
$$= -\mathbb{E}\left( \big(\mathbf{1}_{\mathcal{A}_n} - \mathbf{1}_{\mathcal{A}_n}\mathbf{1}_{\mathcal{A}_{n+1}}\big) h(\theta_{n+1}) \right) - \mathbb{E}\left( \big(\mathbf{1}_{\mathcal{A}_n}\mathbf{1}_{\mathcal{A}_{n+1}} - \mathbf{1}_{\mathcal{A}_{n+1}}\big) h(\theta_{n+1}) \right)$$
$$\leq \frac{1}{\ln\big(\min\{\hat{\delta}, \frac{1}{2}\}\big)} \mathbb{E}\left( \big(\mathbf{1}_{\mathcal{A}_n} - \mathbf{1}_{\mathcal{A}_n}\mathbf{1}_{\mathcal{A}_{n+1}}\big) \right) + \frac{1}{\ln\big(\min\{\hat{\delta}, \frac{1}{2}\}\big)} \mathbb{E}\left( \big(\mathbf{1}_{\mathcal{A}_n}\mathbf{1}_{\mathcal{A}_{n+1}} - \mathbf{1}_{\mathcal{A}_{n+1}}\big) \right)$$
$$= \frac{1}{\ln\big(\min\{\hat{\delta}, \frac{1}{2}\}\big)} \mathbb{E}\big(\mathbf{1}_{\mathcal{A}_n} - \mathbf{1}_{\mathcal{A}_{n+1}}\big). \tag{32}$$

We make the sum of Equation (30), getting

$$\mathbb{E}\left(\mathbf{1}_{\mathcal{A}_{n+1}}h(\theta_{n+1})\right) - \mathbb{E}\left(I_1 h(\theta_1)\right) \leq -\sum_{k=2}^{n}\mathbb{E}\left(\mathbf{1}_{A_k}\frac{\left\|\nabla g(\theta_k)\right\|^2}{\sqrt{S_{k-1}}g(\theta_k)|\ln(g(\theta_k))|^{1+\alpha}}\right)$$

$$+\sum_{k=2}^{n}\mathbb{E}\left(\mathbf{1}_{A_k}\frac{\left(\nabla g(\theta_k)\right)^{\top}\nabla g(\theta_k,\xi_k)}{g(\theta_k)|\ln(g(\theta_k))|^{1+\alpha}}\left(\frac{1}{\sqrt{S_{k-1}}}-\frac{1}{\sqrt{S_k}}\right)\right)$$

$$+\sum_{k=2}^{n}\mathbb{E}\left(\mathbf{1}_{A_k}\frac{T_k\|\nabla g(\theta_k,\xi_k)\|^2}{S_k}\right)+\sum_{k=1}^{n}\mathbb{E}\left(\mathbf{1}_{A_k}\frac{c_0\alpha_0^3\|\nabla g(\theta_k,\xi_k)\|^3}{S_k^{\frac{3}{2}}}\right)$$

$$-\sum_{k=1}^{n}\mathbb{E}\left((\mathbf{1}_{A_k}-\mathbf{1}_{A_{k+1}})h(\theta_{k+1})\right).$$

We can get that

$$\sum_{n=2}^{+\infty}\mathbb{E}\left(\mathbf{1}_{\mathcal{A}_n}\frac{\left\|\nabla g(\theta_n)\right\|^2}{\sqrt{S_{n-1}}g(\theta_n)|\ln(g(\theta_n))|^{1+\alpha}}\right)\leq \mathbb{E}\left(I_1 h(\theta_1)\right)$$

$$+\sum_{k=1}^{n}\mathbb{E}\left(\mathbf{1}_{A_k}\frac{\left(\nabla g(\theta_k)\right)^{\top}\nabla g(\theta_k,\xi_k)}{g(\theta_k)|\ln(g(\theta_k))|^{1+\alpha}}\left(\frac{1}{\sqrt{S_{k-1}}}-\frac{1}{\sqrt{S_k}}\right)\right)$$

$$+\sum_{k=1}^{n}\mathbb{E}\left(\mathbf{1}_{A_k}\frac{T_k\|\nabla g(\theta_k,\xi_k)\|^2}{S_k}\right)+\sum_{k=1}^{n}\mathbb{E}\left(\mathbf{1}_{A_k}\frac{c_0\alpha_0\|\nabla g(\theta_k,\xi_k)\|^3}{S_k^{\frac{3}{2}}}\right) \quad (33)$$

$$-\sum_{k=1}^{n}\mathbb{E}\left((\mathbf{1}_{A_k}-\mathbf{1}_{A_{k+1}})h(\theta_{k+1})\right).$$

For the third term in the right side of Equation (33), we have

$$\sum_{k=1}^{n}\mathbb{E}\left(\mathbf{1}_{A_k}\frac{T_k\|\nabla g(\theta_k,\xi_k)\|^2}{S_k}\right)=\sum_{k=1}^{n}\mathbb{E}\left(\mathbf{1}_{A_k}\mathbf{1}\left(\frac{1}{\sqrt{S_k}}<\tilde{k}g(\theta_k)\right)\frac{d_0\|\nabla g(\theta_k,\xi_k)\|^2}{\|\theta_k\|^{1+\alpha}S_k}\right)$$

$$+\sum_{k=1}^{n}\mathbb{E}\left(\mathbf{1}_{A_k}\mathbf{1}\left(\frac{1}{\sqrt{S_k}}\geq\tilde{k}g(\theta_k)\right)\frac{d_0\|\nabla g(\theta_k,\xi_k)\|^2}{\|\theta_k\|^{1+\alpha}S_k}\right)$$

$$\leq\sum_{k=1}^{n}\mathbb{E}\left(\mathbf{1}_{A_k}\frac{d_0\tilde{k}g(\theta_k)\|\nabla g(\theta_k,\xi_k)\|^2}{\|\theta_k\|^2 S_k}\right)+\sum_{k=1}^{n}\mathbb{E}\left(\frac{4d_0\|\nabla g(\theta_k,\xi_k)\|^2}{S_k\ln^{1+\alpha}S_k}\right).$$

taking proper $\tilde{k}$, we can make

$$\sum_{k=1}^{n}\mathbb{E}\left(\mathbf{1}_{A_k}\frac{T_k\|\nabla g(\theta_k,\xi_k)\|^2}{S_k}\right)$$

$$\leq\frac{T_1\|\nabla g(\theta_1,\xi_1)\|^2}{S_1}+\frac{1}{2}\sum_{k=2}^{n}\mathbb{E}\left(\mathbf{1}_{A_k}\frac{\left\|\nabla g(\theta_k)\right\|^2}{\sqrt{S_{k-1}}g(\theta_k)|\ln(g(\theta_k))|^{1+\alpha}}\right)+\sum_{k=2}^{n}\mathbb{E}\left(\frac{4\hat{d}_0\|\nabla g(\theta_k,\xi_k)\|^2}{S_k|\ln S_k|^{1+\alpha}}\right)$$

$$\leq\frac{1}{2}\sum_{k=2}^{n}\mathbb{E}\left(\mathbf{1}_{A_k}\frac{\left\|\nabla g(\theta_k)\right\|^2}{\sqrt{S_{k-1}}g(\theta_k)|\ln(g(\theta_k)|^{1+\alpha}}\right)+4\hat{d}_0\int_{S_2}^{+\infty}\frac{1}{x|\ln x|^{1+\alpha}}dx+\frac{T_1\|\nabla g(\theta_1,\xi_1)\|^2}{S_1}.$$

$$(34)$$

Substitute Equation (31), Equation (32) and Equation (34) into Equation (33), getting

$$\sum_{n=2}^{+\infty}\mathbb{E}\left(\mathbf{1}_{\mathcal{A}_n}\frac{\left\|\nabla g(\theta_n)\right\|^2}{\sqrt{S_{n-1}}g(\theta_n)|\ln(g(\theta_n))|^{1+\alpha}}\right)<+\infty. \quad (35)$$

For the event $A_n^-:=\{\theta_n\notin S^{(\hat{\delta})}\}$ $(\hat{\delta}=(\ln 2)/N)$. Combining Equation (27), Through Lemma B.4, we have

$$\sum_{k=2}^{n}\mathbb{E}\left(\mathbf{1}_{A_k}^-\frac{\left\|\nabla g(\theta_k)\right\|^2}{\sqrt{S_{k-1}}g(\theta_k)|\ln(g(\theta_k))|^{1+\alpha}}\right)<\tilde{c}_0\sum_{k=2}^{n}\mathbb{E}\left(\frac{\left\|\nabla g(\theta_k)\right\|^2}{\sqrt{S_{k-1}}}\right)<+\infty, \quad (36)$$

where $\tilde{c}_0$ is a constant which can not effect the result. We calculate Equation (35) plus Equation (36), getting

$$\sum_{k=2}^{n} \mathbb{E}\left(\frac{\left\|\nabla g(\theta_k)\right\|^2}{\sqrt{S_{k-1}}g(\theta_k)|\ln(g(\theta_k))|^{1+\alpha}}\right) < +\infty. \tag{37}$$

$\square$

### C.7 Proof of Lemma B.5

*Proof.* We know

$$g(\theta) = -\frac{1}{N}\sum_{i=1}^{N}\left(y_i \ln\left(\frac{1}{1+e^{-\mathrm{sgn}(y_i-0.5)\theta^\top x_i}}\right) + (1-y_i)\ln\left(1-\frac{1}{1+e^{-\mathrm{sgn}(y_i-0.5)\theta^\top x_i}}\right)\right).$$

We defined

$$f(\theta, x_i) := \frac{1}{1+e^{-\mathrm{sgn}(y_i-0.5)\theta^\top x_i}},$$

and

$$f_{x_i}(\theta, x_i) := \begin{cases} f(\theta, x_i), & \text{if } y_i = 0, \\ 1 - f(\theta, x_i), & \text{if } y_i = 1. \end{cases} \tag{38}$$

We can calculate the gradient

$$\nabla g(\theta) = -\frac{1}{N}\sum_{i=1}^{N} f_{x_i}(\theta, x_i)x_i.$$

Then we get

$$-\mathbb{E}\left(\frac{\nabla f(\theta_n)^\top \nabla g(\theta_n, \xi_n)}{\sqrt{S_{n-1}}}\right) = \mathbb{E}\left(\left(\frac{\hat{\theta}^*\|\theta_n\| - \frac{\theta_n\theta_n^\top\hat{\theta}^*}{\|\theta_n\|}}{(\|\theta_n\|+1)^2}\right)^\top \frac{\nabla g(\theta_n)}{\sqrt{S_{n-1}}}\right) + \mathbb{E}\left(\left(\frac{\hat{\theta}^*}{(\|\theta_n\|+1)^2}\right)^\top \frac{\nabla g(\theta_n)}{\sqrt{S_{n-1}}}\right)$$

$$= \mathbb{E}\left(\left(\frac{\hat{\theta}^*\|\theta_n\| - \frac{\theta_n\theta_n^\top\hat{\theta}^*}{\|\theta_n\|}}{(\|\theta_n\|+1)^2}\right)^\top \frac{\nabla g(\theta_n)}{\sqrt{S_{n-1}}}\right) - \mathbb{E}\left(\frac{1}{N\sqrt{S_{n-1}}}\left(\frac{\hat{\theta}^*}{(\|\theta_n\|+1)^2}\right)^\top \sum_{i=1}^{N}\mathrm{sgn}(y_i-0.5)f_{x_i}(\theta_n, x_i)x_i\right)$$

$$\le \mathbb{E}\left(\left(\frac{\hat{\theta}^*\|\theta_n\| - \frac{\theta_n\theta_n^\top\hat{\theta}^*}{\|\theta_n\|}}{(\|\theta_n\|+1)^2}\right)^\top \frac{\nabla g(\theta_n)}{\sqrt{S_{n-1}}}\right)$$

$$= -\mathbb{E}\left(\frac{1}{N\sqrt{S_{n-1}}}\left(\frac{\hat{\theta}^*\|\theta_n\| - \frac{\theta_n\theta_n^\top\hat{\theta}^*}{\|\theta_n\|}}{(\|\theta_n\|+1)^2}\right)^\top \sum_{i=1}^{N}\mathrm{sgn}(y_i-0.5)f_{x_i}(\theta_n, x_i)x_i\right)$$

$$= -\mathbb{E}\left(\frac{1}{N\sqrt{S_{n-1}}}\sum_{i=1}^{N}\mathrm{sgn}(y_i-0.5)f_{x_i}(\theta_n, x_i)\left(\frac{\hat{\theta}^*\|\theta_n\| - \frac{\theta_n\theta_n^\top\hat{\theta}^*}{\|\theta_n\|}}{(\|\theta_n\|+1)^2}\right)^\top x_i\right)$$

$$= \mathbb{E}\left(\frac{1}{N\sqrt{S_{n-1}}}\sum_{i=1}^{N}\mathrm{sgn}(y_i-0.5)f_{x_i}(\theta_n, x_i)\frac{\frac{\theta_n^\top\hat{\theta}^*\theta_n^\top x_i}{\|\theta_n\|} - \hat{\theta}^{*\top}x_i\|\theta_n\|}{(\|\theta_n\|+1)^2}\right)$$

$$= \mathbb{E}\left(\frac{1}{N\sqrt{S_{n-1}}}\sum_{i=1}^{N}\mathrm{sgn}(y_i-0.5)f_{x_i}(\theta_n, x_i)\left(\frac{\theta_n^\top x_i - \hat{\theta}^{*\top}x_i\|\theta_n\|}{(\|\theta_n\|+1)^2} - \frac{\theta_n^\top x_i}{2(\|\theta_n\|+1)^2}\left\|\frac{\theta_n}{\|\theta_n\|} - \hat{\theta}^*\right\|^2\right)\right)$$

$$\le \mathbb{E}\left(\frac{1}{N\sqrt{S_{n-1}}}\sum_{i=1}^{N}\mathrm{sgn}(y_i-0.5)f_{x_i}(\theta_n, x_i)\frac{\theta_n^\top x_i - \hat{\theta}^{*\top}x_i\|\theta_n\|}{(\|\theta_n\|+1)^2}\right) + \beta_n, \tag{39}$$

where

$$\beta_n$$

$$:= \mathbb{E}\left(\mathbf{1}(\theta_n \text{ is not a margin vector})\frac{1}{N\sqrt{S_{n-1}}}\sum_{i=1}^{N}\mathrm{sgn}(y_i-0.5)f_{x_i}(\theta_n, x_i)\frac{|\theta_n^\top x_i|}{2(\|\theta_n\|+1)^2}\left\|\frac{\theta_n}{\|\theta_n\|} - \hat{\theta}^*\right\|^2\right).$$

through Lemma B.10, we know following inequity

$$\beta_n \leq \frac{\max_{1 \leq i \leq N}\{\|x_i\|^2\}}{2} \cdot \mathbb{E}\left(\mathbf{1}(\theta_n \text{ is not a margin vector})\frac{1}{\sqrt{S_{n-1}}}\right)$$

$$\leq \frac{N^2 \max_{1 \leq i \leq N}\{\|x_i\|^2\}}{2k_1^2 \ln^2 2} \cdot \mathbb{E}\left(\mathbf{1}(\theta_n \text{ is not a margin vector})\frac{\|\nabla g(\theta_n)\|^2}{\sqrt{S_{n-1}}}\right)$$

$$\leq \frac{N^2 \max_{1 \leq i \leq N}\{\|x_i\|^2\}}{2k_1^2 \ln^2 2} \cdot \mathbb{E}\left(\frac{\|\nabla g(\theta_n)\|^2}{\sqrt{S_{n-1}}}\right)$$

For convenient, we assign

$$H_n := \frac{1}{N\sqrt{S_{n-1}}}\sum_{i=1}^{N}\psi_i f_{x_i}(\theta_n, x_i)\frac{{\theta_n}^\top x_i - {\theta^*}^\top x_i\|\theta_n\|}{(\|\theta_n\| + 1)^2}, \tag{40}$$

where $\psi_i := \text{sgn}(y_i - 0.5)$. We denote the index of the support vector as $\mathbf{i}_n := \{i|i = \arg\min_{1 \leq i \leq N} \psi_i {\theta_n}^\top x_i/\|\theta_n\|\}$, and $i_n$ is a element of $\mathbf{i}_n$. Then for $H_n$, we have $\exists \hat{k}_0 > 0$, such that

$$H_n = \frac{1}{N\sqrt{S_{n-1}}}\sum_{i=1}^{N}\psi_i f_{x_i}(\theta_n, x_i)\frac{{\theta_n}^\top x_i - {\hat{\theta}^*}^\top x_i\|\theta_n\|}{(\|\theta_n\| + 1)^2}$$

$$= \frac{\|\theta_n\|}{N(\|\theta_n + 1\|)^2\sqrt{S_{n-1}}}\left(\sum_{i \in \mathbf{i}_n}\psi_i f_{x_i}(\theta_n, x_i)\left(\frac{{\theta_n}^\top x_i}{\|\theta_n\|} - {\hat{\theta}^*}^\top x_i\right) + \sum_{i \notin \mathbf{i}_n}\psi_i f_{x_i}(\theta_n, x_i)\left(\frac{{\theta_n}^\top x_i}{\|\theta_n\|} - {\hat{\theta}^*}^\top x_i\right)\right)$$

$$= \frac{f_{x_i}(\theta_n, x_i)\|\theta_n\|}{N(\|\theta_n + 1\|)^2\sqrt{S_{n-1}}}\left(\left(\sum_{i \in \mathbf{i}_n}\psi_i\frac{{\theta_n}^\top x_i}{\|\theta_n\|} - {\hat{\theta}^*}^\top x_i\right) + \sum_{i \notin \mathbf{i}_n}\psi_i\frac{f_{x_i}(\theta_n, x_i)}{f_{x_{i_n}}(\theta_n, x_{i_n})}\left(\frac{{\theta_n}^\top x_i}{\|\theta_n\|} - {\hat{\theta}^*}^\top x_i\right)\right)$$

$$\leq \frac{f_{x_i}(\theta_n, x_i)\|\theta_n\|}{N(\|\theta_n + 1\|)^2\sqrt{S_{n-1}}}\left(\left(\sum_{i \in \mathbf{i}_n}\psi_i\frac{{\theta_n}^\top x_i}{\|\theta_n\|} - {\hat{\theta}^*}^\top x_i\right) + \hat{k}_0\sum_{i \notin \mathbf{i}_n}\frac{\psi_i}{e^{(d_{n,i} - d_{n,i_n})(\|\theta_n\| + 1)}}\left(\frac{{\theta_n}^\top x_i}{\|\theta_n\|} - {\hat{\theta}^*}^\top x_i\right)\right),$$

where $d_{n,i} := |{\theta_n}^\top x_i|/\|\theta_n\|$. Through Lemma B.6, we know that there exists $\hat{\delta}_1 > 0$, $\hat{r} > 0$ making when $\left\|(\theta_n/\|\theta_n\|) - \hat{\theta}^*\right\| < \mathcal{L} := \min\{\hat{\delta}_1, \tilde{\delta}_0\}$, ($\tilde{\delta}_0$ is defined in Lemma B.9) for any $j \neq i_n$, there is

$$\left|\frac{\theta^\top x_j}{\|\theta\|} - \hat{\theta}^* x_j\right| < \hat{r}\left|\frac{\theta^\top x_{i_n}}{\|\theta\|} - \hat{\theta}^* x_{i_n}\right|.$$

We construct two events

$$\mathcal{C}_n^+ := \left\{\left\|\frac{\theta_n}{\|\theta_n\|} - \hat{\theta}^*\right\| \geq \mathcal{L}\right\}, \mathcal{C}_n^- := \left\{\left\|\frac{\theta_n}{\|\theta_n\|} - \hat{\theta}^*\right\| < \mathcal{L}\right\},$$

and their characteristic function as $\mathbf{1}_{\mathcal{C}_n^+}$. Natruely, we can separate $H_n$ as

$$H_n = \mathbf{1}_{\mathcal{C}_n^-}H_n + \mathbf{1}_{\mathcal{C}_n^+}H_n. \tag{41}$$

For $\mathbf{1}_{\mathcal{C}_n^-}H_n$, we have

$$\mathbf{1}_{\mathcal{C}_n^-}H_n \leq \mathbf{1}_{\mathcal{C}_n^-}\frac{f_{x_{i_n}}(\theta_n, x_{i_n})\|\theta_n\|}{N(\|\theta_n\| + 1)^2\sqrt{S_{n-1}}}\left(\sum_{i \in \mathbf{i}_n}\psi_i\left(\frac{{\theta_n}^\top x_i}{\|\theta_n\|} - {\hat{\theta}^*}^\top x_i\right)\right.$$

$$\left. + \mathbf{1}_{\mathcal{C}_n^-}\hat{k}_0\sum_{i \notin \mathbf{i}_n}\frac{\psi_i}{e^{(d_{n,i} - d_{n,i_n})(\|\theta_n\| + 1)}}\left(\frac{{\theta_n}^\top x_i}{\|\theta_n\|} - {\hat{\theta}^*}^\top x_i\right)\right). \tag{42}$$

In Equation (42), we know the first term in the bracket is negative. For the second term, we have

$$\mathbf{1}_{\mathcal{C}_n^-}\sum_{i=1, i \notin \mathbf{i}_n}^{N}\frac{\psi_i}{e^{(d_{n,i} - d_{n,i_n})(\|\theta_n\| + 1)}}\left(\frac{{\theta_n}^\top x_i}{\|\theta_n\|} - {\hat{\theta}^*}^\top x_i\right) = \mathbf{1}_{\mathcal{C}_n^-}\sum_{i=1, i \notin \mathbf{i}_n}^{N}\left(\mathbf{1}\big((d_{n,i} - d_{n,i_n})(\|\theta_n\| + 1) < \hat{U}\big)\right.$$

$$\left. + \mathbf{1}\big((d_{n,i} - d_{n,i_n})(\|\theta_n\| + 1) \geq \hat{U}\big)\right)\frac{\psi_{i_n}}{e^{(d_{n,i} - d_{n,i_n})(\|\theta_n\| + 1)}}\left(\frac{{\theta_n}^\top x_i}{\|\theta_n\|} - {\hat{\theta}^*}^\top x_i\right). \tag{43}$$

where $\hat{U} > 0$ is an undetermined constant. We know where

$$(d_{n,i} - d_{n,i_n})(\|\theta_n\| + 1) < \hat{U},$$

which means

$$\psi_i\left(\frac{\theta_n^\top x_i}{\|\theta_n\|} - \hat{\theta}^{*\top} x_i\right) \leq \mathbf{1}\left(\frac{\theta_n^\top x_i}{\|\theta_n\|} - \hat{\theta}^{*\top} x_i \geq 0\right) \cdot \left(\hat{\theta}^{*\top} x_i - \frac{\theta_n^\top x_i}{\|\theta_n\|}\right)$$

$$\leq \mathbf{1}\left(\frac{\theta_n^\top x_i}{\|\theta_n\|} - \hat{\theta}^{*\top} x_i \geq 0\right) \cdot \left(\frac{\theta_n^\top x_i}{\|\theta_n\|} - \frac{\theta_n^\top x_{i_n}}{\|\theta_n\|} + \frac{\theta_n^\top x_{i_n}}{\|\theta_n\|} - \theta^{*\top} x_{i_n} + \theta^{*\top} x_{i_n} - \hat{\theta}^{*\top} x_i\right)$$

$$\leq (d_{n,i} - d_{n,i_n}) \leq \frac{\hat{U}}{\|\theta_n\| + 1}.$$

On the other hand, Due to the characteristic function $\mathbf{1}_{\mathcal{C}_n^-}$, we can confine Equation (43) on the set $\mathcal{C}_n^-$. That means

$$\mathbf{1}_{\mathcal{C}_n^-} \sum_{i \notin \mathbf{i}_n} \mathbf{1}\left((d_{n,i} - d_{n,i_n})(\|\theta_n\| + 1) < \hat{U}\right) \frac{\psi_i}{e^{(d_{n,i} - d_{n,i_n})(\|\theta_n\| + 1)}}\left(\frac{\theta_n^\top x_i}{\|\theta_n\|} - \hat{\theta}^{*\top} x_i\right)$$

$$\leq \hat{c} N \frac{\hat{U}}{\|\theta_n\| + 1}, \tag{44}$$

and

$$\sum_{i \notin \mathbf{i}_n} \mathbf{1}\left((d_{n,i} - d_{n,i_n})(\|\theta_n\| + 1) \geq \hat{U}\right) \frac{\psi_i}{e^{(d_{n,i} - d_{n,i_n})(\|\theta_n\| + 1)}}\left(\frac{\theta_n^\top x_i}{\|\theta_n\|} - \hat{\theta}^{*\top} x_i\right)$$

$$\leq \frac{N\hat{r}}{e^{\hat{U}}}\left|\frac{\theta_n^\top x_{i_n}}{\|\theta_n\|} - \hat{\theta}^{*\top} x_{i_n}\right|. \tag{45}$$

We substitute Equation (45) and Equation (44) into Equation (43), getting

$$\sum_{i=1, i \notin \mathbf{i}_n}^{N} \frac{\psi_i}{e^{(d_{n,i} - d_{n,i_n})(\|\theta_n\| + 1)}}\left(\frac{\theta_n^\top x_i}{\|\theta_n\|} - \hat{\theta}^{*\top} x_i\right) \leq \hat{k}_0 \hat{c} N \frac{\hat{U}}{\|\theta_n\| + 1}$$

$$+ \hat{k}_0 \frac{N\hat{r}}{e^{\hat{U}}}\left|\frac{\theta_n^\top x_{i_n}}{\|\theta_n\|} - \hat{\theta}^{*\top} x_{i_n}\right|. \tag{46}$$

We substitute Equation (46) into Equation (42), acquiring

$$\mathbf{1}_{\mathcal{C}_n^-} H_n \leq \mathbf{1}_{\mathcal{C}_n^-} \frac{f_{x_{i_n}}(\theta_n, x_{i_n})\|\theta_n\|}{N(\|\theta_n\| + 1)^2 \sqrt{S_{n-1}}}\left(\psi_{i_n}\left(\frac{\theta_n^\top x_{i_n}}{\|\theta_n\|} - \hat{\theta}^{*\top} x_{i_n}\right) + \hat{k}_0 \hat{c} N \frac{\hat{U}}{\|\theta_n\| + 1}\right.$$

$$\left. + \hat{k}_0 \frac{N\hat{r}}{e^{\hat{U}}}\left|\frac{\theta_n^\top x_{i_n}}{\|\theta_n\|} - \hat{\theta}^{*\top} x_{i_n}\right|\right).$$

We take the undetermined constant $\hat{U} = \ln\left(2\hat{k}_0 N\hat{r}\right)$, getting $\exists \tilde{M} > 0$, such that

$$\mathbf{1}_{\mathcal{C}_n^-} H_n \leq \mathbf{1}_{\mathcal{C}_n^-} \frac{f_{x_{i_n}}(\theta_n, x_{i_n})\|\theta_n\|}{N(\|\theta_n\| + 1)^2 \sqrt{S_{n-1}}}\left(\frac{1}{2}\psi_{i_n}\left(\frac{\theta_n^\top x_{i_n}}{\|\theta_n\|} - \hat{\theta}^{*\top} x_{i_n}\right) + \frac{\tilde{M}}{\|\theta_n\| + 1}\right). \tag{47}$$

For $\mathbf{1}_{\mathcal{C}_n^+} H_n$ in Equation (41), we can use the similar techniques (from Equation (42) to Equation (47)) to acquire $\exists \tilde{M}_1 > 0$, such that

$$\mathbf{1}_{\mathcal{C}_n^+} H_n \leq \mathbf{1}_{\mathcal{C}_n^+} \frac{f_{x_{i_n}}(\theta_n, x_{i_n})\|\theta_n\|}{N(\|\theta_n\| + 1)^2 \sqrt{S_{n-1}}}\left(\frac{1}{2}\psi_{i_n}\left(\frac{\theta_n^\top x_{i_n}}{\|\theta_n\|} - \hat{\theta}^{*\top} x_{i_n}\right) + \frac{\tilde{M}_1}{e^{s'\|\theta_n\|}}\right). \tag{48}$$

Then we calculate Equation (48) plus Equation (47), and we get $\exists r_0 > 0, \tilde{M}_0 > 0$, such that

$$H_n \leq \frac{f_{x_{i_n}}(\theta_n, x_{i_n})\|\theta_n\|}{N(\|\theta_n\| + 1)^2 \sqrt{S_{n-1}}}\left(\frac{1}{2}\psi_{i_n}\left(\frac{\theta_n^\top x_{i_n}}{\|\theta_n\|} - \hat{\theta}^{*\top} x_{i_n}\right) + \frac{r_0}{\|\theta_n\| + 1} + \frac{\tilde{M}_0}{e^{s'\|\theta_n\|}}\right). \tag{49}$$

With this, we complete the proof. $\qquad\square$

## C.8 PROOF OF THEOREM 4.1

*Proof.* For the sequence $\{S_n\}$, we separate the proof into two situation. The first situation is $S_n < +\infty$. In this situation, we use Lemma B.4 and Lemma B.2, getting

$$\frac{\|\nabla g(\theta_n)\|^2}{\sqrt{S_{n-1}}} \to 0.$$

Combine $\lim_{n \to +\infty} S_n < +\infty$, getting

$$\|\nabla g(\theta_n)\| \to 0. \tag{50}$$

Then we consider the second situation $S_n \to +\infty$. Through Equation (10) and Equation (16), we can get $g(\theta_{n+1}) - g(\theta_n)$ as

$$g(\theta_{n+1}) - g(\theta_n) \le \hat{\alpha}_0 \frac{1}{\sqrt{S_{n-1}}} + \hat{T}_n, \tag{51}$$

where $\hat{\alpha}_0 > 0$ is a constant and $\hat{T}_n$ is a sequence which satisfies $\sum_{n=1}^{+\infty} \hat{T}_n < +\infty$ *a.s.*. Then we can get

$$
\begin{aligned}
\sum_{n=2}^{+\infty} \frac{1}{\sqrt{S_{n-1}}} &\ge \frac{1}{a} \sum_{n=1}^{+\infty} \frac{\mathbb{E}\left(\|\nabla g(\theta_n, \xi_n)\|^2 \big| \mathscr{F}_n\right) - M_0 \|\nabla g(\theta_n)\|^2}{\sqrt{S_{n-1}}} \\
&\ge \frac{1}{a} \sum_{n=2}^{+\infty} \frac{\mathbb{E}\left(\|\nabla g(\theta_n, \xi_n)\|^2 \big| \mathscr{F}_n\right)}{\sqrt{S_{n-1}}} - \zeta_0 \\
&= \frac{1}{a} \sum_{n=2}^{+\infty} \frac{\|\nabla g(\theta_n, \xi_n)\|^2}{\sqrt{S_{n-1}}} + \frac{1}{a} \sum_{n=2}^{+\infty} \frac{\mathbb{E}\left(\|\nabla g(\theta_n, \xi_n)\|^2 \big| \mathscr{F}_n\right) - \|\nabla g(\theta_n, \xi_n)\|^2}{\sqrt{S_{n-1}}} - \zeta_0,
\end{aligned} \tag{52}
$$

where $\zeta_0 := \sum_{n=2}^{+\infty} M_0 \|\nabla g(\theta_n)\|^2 / a\sqrt{S_{n-1}} < +\infty$ *a.s.*. Next we aim to prove $\sum_{n=2}^{+\infty} 1/\sqrt{S_{n-1}} = +\infty$ *a.s.* by contradiction. We assume $\sum_{n=2}^{+\infty} 1/\sqrt{S_{n-1}} < +\infty$ *a.s.*. Then through Lemma B.2, we get that

$$\frac{1}{a} \sum_{n=2}^{+\infty} \frac{\mathbb{E}\left(\|\nabla g(\theta_n, \xi_n)\|^2 \big| \mathscr{F}_n\right) - \|\nabla g(\theta_n, \xi_n)\|^2}{\sqrt{S_{n-1}}}$$

is convergence *a.s.*. Substitute it into Equation (52), acquiring

$$
\begin{aligned}
&\frac{1}{a} \sum_{n=2}^{+\infty} \frac{\|\nabla g(\theta_n, \xi_n)\|^2}{\sqrt{S_{n-1}}} \\
&\le \sum_{n=2}^{+\infty} \frac{1}{\sqrt{S_{n-1}}} - \frac{1}{a} \sum_{n=2}^{+\infty} \frac{\mathbb{E}\left(\|\nabla g(\theta_n, \xi_n)\|^2 \big| \mathscr{F}_n\right) - \|\nabla g(\theta_n, \xi_n)\|^2}{\sqrt{S_{n-1}}} + \zeta_0 < +\infty.
\end{aligned} \tag{53}
$$

However, we know

$$\frac{1}{a} \sum_{n=2}^{+\infty} \frac{\|\nabla g(\theta_n, \xi_n)\|^2}{\sqrt{S_{n-1}}} > \frac{1}{a} \int_{S_1}^{+\infty} \frac{1}{\sqrt{x}} dx = +\infty.$$

It contradicts with Equation (53). That means

$$\sum_{n=2}^{+\infty} \frac{1}{\sqrt{S_{n-1}}} = +\infty.$$

Combining it with

$$\sum_{n=2}^{+\infty} \frac{\|\nabla g(\theta_n)\|^2}{\sqrt{S_{n-1}}} < +\infty \ \ a.s., \tag{54}$$

we acquire that there is a subsequence $\{\|\nabla g(\theta_{k_n})\|^2\}$ of $\{\|\nabla g(\theta_n)\|^2\}$ which satisfies that

$$\lim_{n \to +\infty} \|\nabla g(\theta_{k_n})\|^2 = 0. \tag{55}$$

Next we aim to prove $\lim_{n \to +\infty} \|\nabla g(\theta_n)\|^2 = 0$. It is equivalent to prove that $\{\|\nabla g(\theta_n)\|^2\}$ has no positive accumulation points, that is to say, $\forall e_0 > 0$, there are only finite values of $\{\|\nabla g(\theta_n)\|\}$ larger than $e_0$. And obviously, we just need to prove $\forall 0 < e_0 < r$, there are only finite values of $\{\|\nabla g(\theta_n)\|\}$ larger than $r$. We prove this by contradiction. We suppose $\exists\, 0 < e < a$, making the set $S = \{\|\nabla g(\theta_n)\|^2 > a\}$ be an infinite set. We assign the *Lipschitz coefficient* of $\nabla g(\theta)$ $(\theta \in \mathbb{R}^d)$ as $c$. Then we assign $b = e/8c$ and define $o = min\{b, e/4\}$. Due to Equation (55), we get there exists a subsequence $\{\theta_{p_n}\}$ of $\{\theta_n\}$ which satisfies $\|\nabla g(\theta_{p_n})\| < o$. We rank $S$ as a subsequence $\{\|\nabla g(\theta_{m_n})\|^2\}$ of $\{\|\nabla g(\theta_n)\|^2\}$. Then there is an infinite subsequence $\{\|\nabla g(\theta_{m_{i_n}})\|^2\}$ of $\{\|\nabla g(\theta_{m_n})\|^2\}$ such that $\forall n \in \mathbb{N}_+, \exists l,\ n_{p_n} \in (m_{i_l}, m_{i_{l+1}})$. For convenient, we abbreviate $\{m_{i_n}\}$ as $\{i_n\}$. And we construct another infinite sequence $\{q_n\}$ as follows

$$q_1 = \max\left\{n : p_1 < n < \min\{m_{i_l : m_{i_l} > p_1}\}, \|\nabla g(\theta_n)\| \le o\right\},$$

$$q_2 = \min\left\{n : n > q_1, \|\nabla g(\theta_n)\| > e\right\},$$

$$q_{2n-1} = \max\left\{n : \min\{m_{i_l} : m_{i_l} > q_{2n-3}\} < n < \min\{m_l : m_l > \min\{m_{i_l} : m_{i_l} > q_{2n-3}\},\right.$$
$$\left.\|\nabla g(\theta_n)\| \le o\right\},$$

$$q_{2n} = \min\left\{n : n > q_{2n-1}, \|\nabla g(\theta_n)\| > e\right\}.$$

Now we prove that $\exists N_0$, when $q_{2n} > N_0$, it has $e < \|\nabla g(\theta_{q_{2n}})\| < r$. The left side is obvious (the definition of $q_{2n}$). And for the right side, we know $\|\nabla g(\theta_{q_{2n}-1})\| \le e$. It follows from Equation (1) that

$$\|\theta_{n+1} - \theta_n\|^2 = \frac{\alpha_0^2}{S_n}\|\nabla g(\theta_n, \xi_n)\|^2$$

$$\le \frac{\alpha_0^2}{S_{n-1}}\left(\|\nabla g(\theta_n, \xi_n)\|^2 - \mathbb{E}\left(\|\nabla g(\theta_n, \xi_n)\|^2 \big| \mathscr{F}_n\right)\right)$$

$$+ \frac{\alpha_0^2}{S_{n-1}}\left(M_0\|\nabla g(\theta_n)\|^2 + a\right).$$

Through previous consequences we can easily find that

$$\sum_{n=2}^{+\infty}\left(\frac{\alpha_0^2}{S_{n-1}}\left(\|\nabla g(\theta_n, \xi_n)\|^2 - \mathbb{E}\left(\|\nabla g(\theta_n, \xi_n)\|^2 \big| \mathscr{F}_n\right)\right) + \frac{\alpha_0^2 M_0\|\nabla g(\theta_n)\|^2}{S_{n-1}}\right)$$
$$< +\infty \ \ a.s..$$

Note that $\alpha_0^2 a / S_{n-1} \to 0, \quad a.s..$ We conclude

$$\|\theta_{n+1} - \theta_n\| \to 0 \ \ a.s.. \tag{56}$$

Then we get $\left|\|\nabla g(\theta_{n+1})\|^2 - \|\nabla g(\theta_n)\|^2\right| \le \left|\|\nabla g(\theta_{n+1})\| - \|\nabla g(\theta_n)\|\right|^2 \le \|\nabla g(\theta_{n+1}) - \nabla g(\theta_n)\|^2 \le c\|\theta_{n+1} - \theta_n\| \to 0 \ \ a.s..$ Then through Lemma B.7, we get

$$\|\nabla g(\theta_n)\|^2 \le 2cg(\theta_n)\ (n \in [q_{2n-1}, q_{2n}]).$$

Then we get

$$e - o < \|\nabla g(\theta_{q_{2n}})\|^2 - \|\nabla g(\theta_{q_{2n-1}})\|^2 < 2cg(\theta_{q_{2n}}) - \|\nabla g(\theta_{q_{2n-1}})\|^2$$

$$= \left(2c\sum_{i=0}^{q_{2n}-q_{2n-1}-1} g(\theta_{q_{2n-1}+i+1}) - g(\theta_{q_{2n-1}+i})\right) + 2cg(\theta_{q_{2n-1}}) - \|\nabla g(\theta_{q_{2n-1}})\|^2.$$

From Equation (51), we obtain

$$g(\theta_{q_{2n-1}+i+1}) - g(\theta_{q_{2n-1}+i}) \le \hat{\alpha}_0 \frac{1}{\sqrt{S_{q_{2n-1}+i}}} + \hat{T}_n.$$

So there is

$$e - o < \sum_{i=0}^{q_{2n} - q_{2n-1} - 1} \frac{\hat{\alpha}_0}{\sqrt{S_{q_{2n-1}+i}}} + \sum_{i=0}^{q_{2n} - q_{2n-1} - 1} \hat{T}_{q_{2n-1}+i} \tag{57}$$
$$+ 2cg(\theta_{q_{2n-1}}) - \left\| \nabla g(\theta_{q_{2n-1}}) \right\|^2.$$

Due to $\|\nabla g(\theta_{q_{2n-1}})\|^2 < o < b$, so we get that $g(\theta_{q_{2n-1}}) < e/8c$. Substitute it into Equation (57). We get

$$\sum_{i=0}^{q_{2n} - q_{2n-1} - 1} \frac{1}{\sqrt{S_{q_{2n-1}+i}}} > \hat{\alpha}_0 - \sum_{i=0}^{q_{2n} - q_{2n-1} - 1} \hat{T}_{q_{2n-1}+i}. \tag{58}$$

Due to $\sum_{n=1}^{+\infty} \hat{T}_n$ is convergence almost surely. So we get that $\sum_{i=0}^{q_{2n} - q_{2n-1} - 1} \hat{T}_{q_{2n-1}+i} \to 0$ $a.s.$ by *Cauchy's test for convergence*. Combining $1/\sqrt{S_{q_{2n-1}+i}} \to 0$ $a.s.$, we get

$$\sum_{i=1}^{q_{2n} - q_{2n-1} - 1} \frac{1}{\sqrt{S_{q_{2n-1}+i}}} > \hat{\alpha}_0 - \frac{1}{\sqrt{S_{q_{2n-1}}}} - \sum_{i=0}^{q_{2n} - q_{2n-1} - 1} \hat{T}_{q_{2n-1}+i} \to \frac{\hat{\alpha}_0}{2} \quad a.s., \tag{59}$$

so there is

$$\sum_{n=1}^{+\infty} \left( \sum_{i=1}^{q_{2n} - q_{2n-1} - 1} \frac{1}{\sqrt{S_{q_{2n-1}+i}}} \right) = +\infty \quad a.s.. \tag{60}$$

But on the other hand, we know $\|\nabla g(\theta_{q_{2n-1}+i})\| > o$ $(i > 0)$. Together with Equation (54), we get

$$\sum_{n=1}^{+\infty} \left( \sum_{i=1}^{q_{2n} - q_{2n-1} - 1} \frac{1}{\sqrt{S_{q_{2n-1}+i}}} \right) < \frac{1}{o} \sum_{n=1}^{+\infty} \left( \sum_{i=1}^{q_{2n} - q_{2n-1} - 1} \frac{\left\| \nabla g(\theta_{q_{2n-1}+i}) \right\|^2}{\sqrt{S_{q_{2n-1}+i}}} \right)$$
$$< \frac{1}{o} \sum_{n=3}^{n} \frac{\left\| \nabla g(\theta_n) \right\|^2}{\sqrt{S_{n-1}}} < +\infty \quad a.s.. \tag{61}$$

It contradicts with Equation (60), so we get that $\|\nabla g(\theta_n)\| \to 0$ $a.s.$. Combining Equation (50), we get $\|\nabla g(\theta_n)\| \to 0$ no matter $S_n < +\infty$ $a.s.$ or $S_n = +\infty$. Through Lemma B.10 and Lemma B.8, we get $g(\theta_n) \to 0$ $a.s.$. In the case of linear separable data set, $g(\theta n) \to 0$ $a.s.$ implies $\|\theta_n\| \to +\infty$ $a.s.$. $\qquad\square$

### C.9 PROOF OF THEOREM 4.2

*Proof.* We assign $\hat{\theta}^* := \theta^*/\|\theta^*\|$. Then, we assign

$$f(\theta) := 1 - \frac{\theta^\top \hat{\theta}^*}{\|\theta\| + 1}. \tag{62}$$

Then we use the *taylor expansion* on $f(\theta_{n+1}) - f(\theta_n)$, getting

$$f(\theta_{n+1}) - f(\theta_n) \leq \nabla f(\theta_n)^\top (\theta_{n+1} - \theta_n) + T_n \|\theta_{n+1} - \theta_n\|^2$$
$$= -\frac{\alpha_0 \nabla f(\theta_n)^\top \nabla g(\theta_n, \xi_n)}{\sqrt{S_n}} + \frac{T_n \alpha_0^2 \|\nabla g(\theta_n, \xi_n)\|^2}{S_n}$$
$$\leq -\frac{\alpha_0 \nabla f(\theta_n)^\top \nabla g(\theta_n, \xi_n)}{\sqrt{S_{n-1}}} + \frac{\alpha_0 \hat{\theta}^{*\top} \nabla g(\theta_n, \xi_n)}{(\|\theta_n\| + 1)^2 \sqrt{S_{n-1}}} \tag{63}$$
$$+ \left( \frac{\hat{\theta}^* \|\theta_n\| - \frac{\theta_n \theta_n^\top \hat{\theta}^*}{\|\theta_n\|}}{(\|\theta_n\| + 1)^2} \right)^\top \alpha_0 \nabla g(\theta_n, \xi_n) \left( \frac{1}{\sqrt{S_{n-1}}} - \frac{1}{\sqrt{S_n}} \right) + \frac{T_n \alpha_0^2 \|\nabla g(\theta_n, \xi_n)\|^2}{S_n},$$

where

$$T_n := \hat{c}_0 \left( \frac{1}{(\|\theta_n\| + 1)^2} + \frac{1}{(\|\theta_{n+1}\| + 1)^2} \right),$$

where $\hat{c}_0$ is a constant which can not effect the result. For convenience, we assign

$$G_n := \left| \left( \frac{\hat{\theta}^* \|\theta_n\| - \frac{\theta_n \theta_n^\top \hat{\theta}^*}{\|\theta_n\|}}{(\|\theta_n\| + 1)^2} \right)^\top \alpha_0 \nabla g(\theta_n, \xi_n) \left( \frac{1}{\sqrt{S_{n-1}}} - \frac{1}{\sqrt{S_n}} \right) \right| + \frac{T_n \alpha_0^2 \|\nabla g(\theta_n, \xi_n)\|^2}{S_n}$$

$$+ \frac{\alpha_0 \hat{\theta}^{*\top} \nabla g(\theta_n, \xi_n)}{(\|\theta_n\| + 1)^2 \sqrt{S_{n-1}}} + \frac{N^2 \max_{1 \le i \le N}\{\|x_i\|^2\}}{2k_1^2 \ln^2 2} \cdot \frac{\|\nabla g(\theta_n)\|^2}{\sqrt{S_{n-1}}}.$$

Then we make the mathematical expectation of Equation (63), getting

$$\mathbb{E}\left(f(\theta_{n+1})\right) - \mathbb{E}\left(f(\theta_n)\right) \le \alpha_0 \mathbb{E}\left(H_n\right) + \mathbb{E}\left(G_n\right), \tag{64}$$

where $H_n$ is defined in Equation (40). Then we make a sum of Equation (64), acquiring

$$\alpha_0 \sum_{k=2}^n \mathbb{E}\left(-H_k\right) \le \mathbb{E}\left(f(\theta_2)\right) + \sum_{k=2}^n \mathbb{E}\left(G_k\right).$$

obviously,

$$\sum_{k=2}^{+\infty} \left\| \mathbb{E}\left(G_k\right) \right\| \le \sum_{k=2}^{+\infty} \mathbb{E}\left\| G_k \right\|$$

$$\le \sum_{k=2}^{+\infty} \mathbb{E} \left\| \left( \frac{\hat{\theta}^* \|\theta_n\| - \frac{\theta_n \theta_n^\top \hat{\theta}^*}{\|\theta_n\|}}{(\|\theta_n\| + 1)^2} \right)^\top \alpha_0 \nabla g(\theta_n, \xi_n) \left( \frac{1}{\sqrt{S_{n-1}}} - \frac{1}{\sqrt{S_n}} \right) \right\| + \sum_{n=2}^{+\infty} \mathbb{E}\left( \frac{T_n \alpha_0^2 \|\nabla g(\theta_n, \xi_n)\|^2}{S_n} \right)$$

$$+ \sum_{n=2}^{+\infty} \mathbb{E}\left( \frac{\alpha_0 \hat{\theta}^{*\top} \nabla g(\theta_n, \xi_n)}{(\|\theta_n\| + 1)^2 \sqrt{S_{n-1}}} \right) + \frac{N \max_{1 \le i \le N}\{\|x_i\|^2\}}{4c \ln 2} \cdot \sum_{n=2}^{+\infty} \mathbb{E}\left( \frac{\|\nabla g(\theta_n)\|^2}{\sqrt{S_{n-1}}} \right), \tag{65}$$

and for the first term on the right side of the above inequality, we have $\exists \hat{T} > 0$, such that

$$\sum_{k=2}^{+\infty} \mathbb{E} \left\| \left( \frac{\hat{\theta}^* \|\theta_n\| - \frac{\theta_n \theta_n^\top \hat{\theta}^*}{\|\theta_n\|}}{(\|\theta_n\| + 1)^2} \right)^\top \alpha_0 \nabla g(\theta_n, \xi_n) \left( \frac{1}{\sqrt{S_{n-1}}} - \frac{1}{\sqrt{S_n}} \right) \right\|$$

$$\le \hat{T} \sum_{k=2}^{+\infty} \mathbb{E}\left( \|\nabla g(\theta_n, \xi_n)\| \left( \frac{1}{\sqrt{S_{n-1}}} - \frac{1}{\sqrt{S_n}} \right) \right).$$

Through Assumption 3.1 and Lemma B.8, we can get that $\exists \hat{T}_0 > 0$, $\hat{T}_1 > 0$, such that

$$\hat{T} \sum_{n=2}^{+\infty} \mathbb{E}\left( \|\nabla g(\theta_n, \xi_n)\| \left( \frac{1}{\sqrt{S_{n-1}}} - \frac{1}{\sqrt{S_n}} \right) \right)$$

$$\le \hat{T} \sum_{n=2}^{+\infty} \mathbb{E}\left( I\left(\|\nabla g(\theta_n)\| \le \delta_0\right) \|\nabla g(\theta_n, \xi_n)\| \left( \frac{1}{\sqrt{S_{n-1}}} - \frac{1}{\sqrt{S_n}} \right) \right)$$

$$+ \hat{T} \sum_{n=2}^{+\infty} \mathbb{E}\left( I\left(\|\nabla g(\theta_n)\| > \delta_0\right) \|\nabla g(\theta_n, \xi_n)\| \left( \frac{1}{\sqrt{S_{n-1}}} - \frac{1}{\sqrt{S_n}} \right) \right) \tag{66}$$

$$\le \hat{T}_0 \frac{1}{\sqrt{S_1}} + \hat{T}_1 \sum_{n=2}^{+\infty} \mathbb{E}\left( \frac{\|\nabla g(\theta_n)\|^2}{\sqrt{S_{n-1}}} \right) < +\infty.$$

For the second term in Equation (65), Through Assumption 3.1 and Lemma 4.2, we have

$$\sum_{n=2}^{+\infty} \mathbb{E}\left( \frac{T_n \alpha_0^2 \|\nabla g(\theta_n, \xi_n)\|^2}{S_n} \right)$$

$$\le \hat{c}_0 \left( \sum_{n=1}^{+\infty} \mathbb{E}\left( \frac{\alpha_0^2 \|\nabla g(\theta_n, \xi_n)\|^2}{(\|\theta_n\| + 1)^2 S_n} \right) + \sum_{n=1}^{+\infty} \mathbb{E}\left( \frac{\alpha_0^2 \|\nabla g(\theta_n, \xi_n)\|^2}{(\|\theta_{n+1}\| + 1)^2 S_n} \right) \right) < +\infty.$$

For the third and forth term of (65), we can use Lemma B.4 and Lemma 4.1 to prove their are convergence. That means $\exists \hat{K}_1 > 0$, such that

$$\sum_{k=1}^{n} \mathbb{E}\left(-H_k\right) \leq \hat{K}_1 < +\infty. \tag{67}$$

Through (49), we get

$$H_n \leq \frac{f_{x_{i_n}}(\theta_n, x_{i_n})\|\theta_n\|}{N(\|\theta_n\|+1)^2\sqrt{S_{n-1}}}\left(\frac{1}{2}\psi_{i_n}\left(\frac{\theta_n^{\top}x_{i_n}}{\|\theta_n\|} - \hat{\theta}^{*\top}x_{i_n}\right) + \frac{r_0}{\|\theta_n\|+1} + \frac{\tilde{M}_0}{e^{s'\|\theta_n\|}}\right). \tag{68}$$

Substitute Equation (68) into Equation (67), we getting

$$\sum_{n=2}^{+\infty} \mathbb{E}\left(\frac{\psi_{i_n}\|\theta_n\|f_{x_{i_n}}(\theta_n, x_{i_n})}{N(\|\theta_n\|+1)^2\sqrt{S_{n-1}}}\left(\hat{\theta}^{*\top}x_{i_n} - \frac{\theta_n^{\top}x_{i_n}}{\|\theta_n\|}\right)\right)$$

$$\leq 2\hat{K}_1 + 2\sum_{n=2}^{+\infty} \mathbb{E}\left(\mathbf{1}_{\mathcal{C}_n^-}\frac{\|\theta_n\|f_{x_{i_n}}(\theta_n, x_{i_n})}{N(\|\theta_n\|+1)^2\sqrt{S_{n-1}}}\cdot\left(\frac{r_0}{\|\theta_n\|+1}\right)\right) + \sum_{n=2}^{+\infty} \mathbb{E}\left(\frac{\|\theta_n\|f_{x_{i_n}}(\theta_n, x_{i_n})}{N(\|\theta_n\|+1)^2\sqrt{S_{n-1}}}\cdot\frac{\hat{M}_0}{e^{s'\|\theta_n\|}}\right).$$

Then we calculate the third term of Equation (64). We know when $\theta_n \in A_n^-$, there is

$$\frac{\|\theta_n\|}{\cdot(\|\theta_n\|+1)^3} \leq \tilde{k}_0\frac{1}{\ln^2(g(\theta_n))} = \tilde{k}_0\frac{1}{|\ln(g(\theta_n))|^{1+1}},$$

where $\tilde{k}_0$ is a constant where can not effect the result. Combine Lemma 4.2, We can get

$$\sum_{n=2}^{+\infty} \mathbb{E}\left(\mathbf{1}_{\mathcal{C}_n^-}\frac{f_{x_{i_n}}(\theta_n, x_{i_n})}{N\|\theta_n\|\sqrt{S_{n-1}}}\cdot\left(\frac{r_0}{\|\theta_n\|+1}\right)\right) \leq \tilde{k}_1\sum_{n=2}^{+\infty} \mathbb{E}\left(\frac{g(\theta_n)}{\sqrt{S_{n-1}}|\ln(g(\theta_n))|^{1+1}}\right) < +\infty.$$

Similarly, through Lemma 4.2, we can get

$$\sum_{n=2}^{+\infty} \mathbb{E}\left(\frac{\|\theta_n\|f_{x_{i_n}}(\theta_n, x_{i_n})}{N(\|\theta_n\|+1)^2\sqrt{S_{n-1}}}\cdot\frac{\hat{M}_0}{e^{s'\|\theta_n\|}}\right) < +\infty.$$

That means we can get

$$\sum_{n=2}^{+\infty} \mathbb{E}\left(\frac{\psi_{i_n}\|\theta_n\|f_{x_{i_n}}(\theta_n, x_{i_n})}{N(1+\|\theta_n\|)^2\sqrt{S_{n-1}}}\left(\hat{\theta}^{*\top}x_{i_n} - \frac{\theta_n^{\top}x_{i_n}}{\|\theta_n\|}\right)\right) < +\infty,$$

We simplify the above inequality and obtain

$$\sum_{n=2}^{+\infty} \mathbb{E}\left(\frac{\|\theta_n\|f_{x_{i_n}}(\theta_n, x_{i_n})}{N(\|\theta_n\|+1)^2\sqrt{S_{n-1}}}\left|\hat{\theta}^{*\top}x_{i_n} - \frac{\theta_n^{\top}x_{i_n}}{\|\theta_n\|}\right|\right) < +\infty.$$

Through Lemma B.1, we have

$$\sum_{n=2}^{+\infty}\frac{\|\theta_n\|f_{x_{i_n}}(\theta_n, x_{i_n})}{N(\|\theta_n\|+1)^2\sqrt{S_{n-1}}}\left|\hat{\theta}^{*\top}x_{i_n} - \frac{\theta_n^{\top}x_{i_n}}{\|\theta_n\|}\right| < +\infty \ \ a.s.. \tag{69}$$

We back to Equation (63). We make a sum of Equation (63), which obtains

$$f(\theta_{n+1}) = f(\theta_1) + \sum_{k=1}^{n}\left(\frac{\hat{\theta}^*(\|\theta_k\|+1) - \frac{\theta_k\theta_k^{\top}\hat{\theta}^*}{\|\theta_k\|}}{(\|\theta_k\|+1)^2}\right)^{\top}\frac{\alpha_0\nabla g(\theta_k, \xi_k)}{\sqrt{S_k}}$$

$$+ \sum_{k=1}^{n}\frac{\hat{c}\alpha_0^2\|\nabla g(\theta_n, \xi_n)\|^2}{S_n}. \tag{70}$$

For the first series sum, we have

$$\sum_{k=2}^{n} \left( \frac{\hat{\theta}^*(\|\theta_k\|+1) - \frac{\theta_k \theta_k^\top \hat{\theta}^*}{\|\theta_k\|}}{(\|\theta_k\|+1)^2} \right)^\top \frac{\alpha_0 \nabla g(\theta_k, \xi_k)}{\sqrt{S_k}} = \sum_{k=2}^{n} \left( \frac{\hat{\theta}^*(\|\theta_k\|+1) - \frac{\theta_k \theta_k^\top \hat{\theta}^*}{\|\theta_k\|}}{(\|\theta_k\|+1)^2} \right)^\top \frac{\alpha_0 \nabla g(\theta_k)}{\sqrt{S_{k-1}}}$$

$$- \sum_{k=2}^{n} \left( \frac{\hat{\theta}^*(\|\theta_k\|+1) - \frac{\theta_k \theta_k^\top \hat{\theta}^*}{\|\theta_k\|}}{(\|\theta_k\|+1)^2} \right)^\top \left( \frac{\alpha_0 \nabla g(\theta_k, \xi_k)}{\sqrt{S_{k-1}}} - \frac{\alpha_0 \nabla g(\theta_k, \xi_k)}{\sqrt{S_k}} \right) + \sum_{k=2}^{n} \zeta_k,$$

where $\{\zeta_n\}$ is a martingale difference sequence. Through Equation (66), Equation (69) and Lemma B.2, we can get

$$\sum_{k=2}^{n} \left( \frac{\hat{\theta}^*(\|\theta_k\|+1) - \frac{\theta_k \theta_k^\top \hat{\theta}^*}{\|\theta_k\|}}{(\|\theta_k\|+1)^2} \right)^\top \frac{\alpha_0 \nabla g(\theta_k, \xi_k)}{\sqrt{S_k}}$$

convergence a.s.. Meanwhile, through Equation (69), we have

$$\sum_{k=2}^{n} \left| \frac{\hat{c}\alpha_0^2 \|\nabla g(\theta_k, \xi_k)\|^2}{S_k} \right| \leq \sum_{k=2}^{n} \left( \frac{1}{\|\theta_k\|^2} + \frac{1}{\|\theta_{k+1}\|^2} \right) \frac{\hat{c}_0 \alpha_0^2 \|\nabla g(\theta_k, \xi_k)\|^2}{S_k} < +\infty \ a.s..$$

That means

$$\sum_{k=1}^{n} \frac{\hat{c}\alpha_0^2 \|\nabla g(\theta_k, \xi_k)\|^2}{S_k}$$

is absolute convergence a.s.. Naturally, it is convergence a.s.. Until now, we already prove two series sums in Equation (70) are both convergence a.s.. That means $f(\theta_n)$ is convergence a.s.. We assign

$$c := \lim_{n \to +\infty} f(\theta_n) \ a.s.,$$

where $c < +\infty$ is a random variable about the trajectory. Through Theorem 4.1, we know

$$\|\theta_n\| \to +\infty.$$

That means

$$\lim_{n \to +\infty} \frac{\|\theta_n\|+1}{\|\theta_n\|} = 1 \ a.s..$$

$$\lim_{n \to +\infty} 1 - \frac{\theta^\top \hat{\theta}^*}{\|\theta\|} = \lim_{n \to +\infty} f(\theta_n) = c,$$

so we can get that

$$\lim_{n \to +\infty} \left\| \frac{\theta_n}{\|\theta_n\|} - \hat{\theta}^* \right\|^2 = 2c \ a.s.$$

Next we aim to prove that $c = 0$ by contradiction. We assume $c > c' > 0$. Then we can conclude that

$$\lim_{n \to +\infty} \left| \hat{\theta}^{*\top} x_{i_n} - \frac{\theta_n^\top x_{i_n}}{\|\theta_n\|} \right| > r(c') > 0 \ a.s..$$

We can further conclude that

$$\sum_{n=2}^{+\infty} \frac{\|\theta_n\| f_{x_{i_n}}(\theta_n, x_{i_n})}{N(\|\theta_n\|+1)^2 \sqrt{S_{n-1}}} \left| \hat{\theta}^{*\top} x_{i_n} - \frac{\theta_n^\top x_{i_n}}{\|\theta_n\|} \right| > r(c') \sum_{n=2}^{+\infty} \frac{f_{x_{i_n}}(\theta_n, x_{i_n})}{N\|\theta_n\| \sqrt{S_{n-1}}} \ a.s.. \tag{71}$$

We can get

$$\sum_{n=2}^{+\infty} \frac{f_{x_{i_n}}(\theta_n, x_{i_n})}{N\|\theta_n\| \sqrt{S_{n-1}}} > \hat{r}' \sum_{n=2}^{+\infty} \frac{\|\theta_n\| f_{x_{i_n}}(\theta_n, x_{i_n})}{N(\|\theta_n\|+1)^2 \sqrt{S_{n-1}}},$$

where $\hat{r}' > 0$ is a constant. Then we can get

$$\sum_{n=2}^{+\infty} \frac{\|\theta_n\| f_{x_{i_n}}(\theta_n, x_{i_n})}{N(\|\theta_n\|+1)^2 \sqrt{S_{n-1}}} > q_1 \sum_{n=1}^{+\infty} \left( \ln \|\theta_{n+1}\| - \ln \|\theta_n\| \right) - q_2 \sum_{n=1}^{+\infty} \frac{\|\nabla g(\theta_n, \xi_n)\|^2}{\|\theta_n\|^2 S_n} = +\infty \ a.s.,$$

where $q_1 > 0$ and $q_2$ are two constants will can not effect the result. That means

$$\sum_{n=2}^{+\infty} \frac{\|\theta_n\| f_{x_{i_n}}(\theta_n, x_{i_n})}{N(\|\theta_n\| + 1)^2 \sqrt{S_{n-1}}} \left| \hat{\theta}^{*\top} x_{i_n} - \frac{\theta_n^\top x_{i_n}}{\|\theta_n\|} \right| = +\infty,$$

which is contradict with Equation (70). That means

$$\lim_{n \to +\infty} \left\| \frac{\theta_n}{\|\theta_n\|} - \hat{\theta}^* \right\| = 0 \ \ a.s.,$$

that is

$$\frac{\theta_n}{\|\theta_n\|} \to \frac{\theta^*}{\|\theta^*\|} \ \ a.s..$$

With this, we complete the proof.

$\square$

## C.10 PROOF OF THEOREM 4.3

*Proof.* For any $0 < \alpha < 1$, we construct a function $r(\theta) := \|\theta\|^\alpha \cdot f(\theta)$ $(0 < \alpha < 1)$, where $f$ is defined in Equation (62). Then we calculate $\nabla r(\theta)$, which obtains

$$\nabla r(\theta) = \nabla(\|\theta\|^\alpha)^\top f(\theta) + (\nabla f(\theta))^\top \|\theta\|^\alpha = \frac{\alpha \frac{\theta}{\|\theta\|} \cdot f(\theta)}{\|\theta\|^{1-\alpha}} + \|\theta\|^\alpha \nabla f(\theta),$$

and $\|\nabla^2 r(\theta)\| = O((\|\theta\| + 1)^{\alpha-2})$. Meanwhile, we assign the Lipschitz constant of $\nabla^2 r(\theta)$ as $c_1$. Subsequently, we get

$$r(\theta_{n+1}) - r(\theta_n) \leq \nabla r(\theta_n)^\top (\theta_{n+1} - \theta_n) + \|\nabla^2 r(\theta_n)\| \cdot \|\theta_{n+1} - \theta_n\|^2 + c_1 \|\theta_{n+1} - \theta_n\|^3$$

$$\leq -\alpha \frac{\alpha_0 (\frac{\theta_n}{\|\theta_n\|})^\top \nabla g(\theta_n, \xi_n) f(\theta_n)}{\sqrt{S_n} \|\theta_n\|^{1-\alpha}} - \|\theta_n\|^\alpha \frac{\alpha_0 \nabla f(\theta_n)^\top \nabla g(\theta_n, \xi_n)}{\sqrt{S_n}} + q_0 \frac{\alpha_0^2 \|\nabla g(\theta_n, \xi_n)\|^2}{(\|\theta_n\| + 1)^{2-\alpha} S_n}$$

$$+ c_1 \alpha_0^3 \frac{\|\nabla g(\theta_n, \xi_n)\|^2}{S_n^3}.$$

$$(72)$$

Notice that

$$\nabla f(\theta_n) = \frac{\theta - \hat{\theta}^* \|\theta_n\|}{(\|\theta_n\| + 1)^2} - \frac{\theta}{2(\|\theta\| + 1)^2} \left\| \frac{\theta_n}{\|\theta_n\|} - \hat{\theta}^* \right\|^2 - \frac{\hat{\theta}^*}{(\|\theta_n\| + 1)^2}.$$

For the first term and second term in the right-hand of Equation (72), we know that

$$- \mathbb{E} \left( \alpha \frac{\alpha_0 (\frac{\theta_n}{\|\theta\|})^\top \nabla g(\theta_n, \xi_n) f(\theta_n)}{\sqrt{S_n} \|\theta_n\|^{1-\alpha}} + \|\theta_n\|^\alpha \frac{\alpha_0 \nabla f(\theta_n)^\top \nabla g(\theta_n, \xi_n)}{\sqrt{S_n}} \Big| \mathscr{F}_n \right)$$

$$\leq -\alpha \frac{\alpha_0 (\frac{\theta_n}{\|\theta_n\|})^\top \nabla g(\theta_n) f(\theta_n)}{\sqrt{S_n} \|\theta_n\|^{1-\alpha}} + \|\theta_n\|^\alpha H_n + \|\theta_n\|^\alpha \frac{\alpha_0}{\sqrt{S_n}} \frac{\|\theta_n\|^2}{2(\|\theta_n\| + 1)^2} \cdot \frac{\theta_n^\top \nabla g(\theta_n)}{\|\theta_n\|^2} \left\| \frac{\theta_n}{\|\theta_n\|} - \hat{\theta}^* \right\|^2,$$

where $H_n$ is defined in Equation (40). Through Theorem 4.2, we know that the vector $\theta_n / \|\theta_n\|$ approaches to the max-margin vector almost surely, which means

$$\frac{\theta_n^\top \nabla g(\theta_n)}{\|\theta_n\|^2} < 0$$

when $n$ is sufficiently large. Then,

$$
- \mathbb{E}\left( \alpha \frac{\alpha_0 (\frac{\theta_n}{\|\theta\|})^\top \nabla g(\theta_n, \xi_n) f(\theta_n)}{\sqrt{S_n} \|\theta_n\|^{1-\alpha}} + \|\theta_n\|^\alpha \frac{\alpha_0 \nabla f(\theta_n)^\top \nabla g(\theta_n, \xi_n)}{\sqrt{S_n}} \bigg| \mathscr{F}_n \right)
$$

$$
\leq (1-\alpha) \frac{\alpha_0 (\frac{\theta_n}{\|\theta\|})^\top \nabla g(\theta_n) f(\theta_n)}{\sqrt{S_n} \|\theta_n\|^{1-\alpha}} + \|\theta_n\|^\alpha H_n + \|\theta_n\|^\alpha \frac{\alpha_0}{\sqrt{S_n}} \left| 1 - \frac{\|\theta_n\|^2}{(\|\theta_n\|+1)^2} \right| \cdot \frac{|\theta_n^\top \nabla g(\theta_n)|}{\|\theta_n\|^2} |f(\theta_n)|
$$

$$
+ \|\theta_n\|^\alpha \frac{\alpha_0}{\sqrt{S_n}} \frac{\|\theta_n\|^2}{(\|\theta_n\|+1)^2} \cdot \frac{|\theta_n^\top \nabla g(\theta_n)|}{\|\theta_n\|^2} \left| f(\theta_n) - \frac{1}{2} \left\| \frac{\theta_n}{\|\theta_n\|} - \hat{\theta}^* \right\|^2 \right|
$$

$$
\leq \|\theta_n\|^\alpha H_n + O\left( \frac{\|\nabla g(\theta_n)\|^2}{\sqrt{S_{n-1}} g(\theta_n) \ln^{2-\alpha}(g(\theta_n))} \right).
$$

Through Equation (49), we have

$$
\|\theta_n\|^\alpha H_n = O\left( \frac{\|\nabla g(\theta_n)\|^2}{\sqrt{S_{n-1}} g(\theta_n) \ln^{2-\alpha}(g(\theta_n))} \right).
$$

Then we use Lemma 4.2 and obtain

$$
\sum_{n=1}^{+\infty} - \mathbb{E}\left( \alpha \frac{\alpha_0 (\frac{\theta_n}{\|\theta\|})^\top \nabla g(\theta_n, \xi_n) f(\theta_n)}{\sqrt{S_n} \|\theta_n\|^{1-\alpha}} + \|\theta_n\|^\alpha \frac{\alpha_0 \nabla f(\theta_n)^\top \nabla g(\theta_n, \xi_n)}{\sqrt{S_n}} \bigg| \mathscr{F}_n \right)
$$

$$
< O\left( \sum_{n=1}^{+\infty} \frac{\|\nabla g(\theta_n)\|^2}{\sqrt{S_{n-1}} g(\theta_n) \ln^{2-\alpha}(g(\theta_n))} \right) < +\infty \ a.s.. \tag{73}
$$

For the third term in the right-hand side of Equation (72), we have $\exists\, Q_1 > 0$, such that

$$
\sum_{n=1}^{+\infty} \frac{\alpha_0^2 \|\nabla g(\theta_n, \xi_n)\|^2}{(\|\theta_n\|+1)^{2-\alpha} S_n} \leq Q_1 \sum_{n=1}^{+\infty} \frac{\|\nabla g(\theta_n, \xi_n)\|^2}{\ln^{2-\alpha}(g(\theta_n)) S_n}
$$

$$
\leq Q_1 \sum_{n=1}^{+\infty} \frac{\|\nabla g(\theta_n, \xi_n)\|^2}{\ln^{2-\alpha}(S_n) S_n} + Q_1 \sum_{n=1}^{+\infty} \frac{\|\nabla g(\theta_n, \xi_n)\|^2 g(\theta_n)}{\ln^{2-\alpha}(g(\theta_n)) \sqrt{S_n}}. \tag{74}
$$

For the fourth term in the right-hand side of Equation (72), we know

$$
\sum_{n=1}^{+\infty} c_1 \alpha_0^3 \frac{\|\nabla g(\theta_n, \xi_n)\|^2}{S_n^3} < +\infty \ a.s.. \tag{75}
$$

Substitute Equation (73), Equation (74) and Equation (75) into Equation (72), we get

$$
\sum_{n=1}^{+\infty} \left( \mathbb{E}\left( r(\theta_{n+1}) \big| \mathscr{F}_n \right) - r(\theta_n) \right) < +\infty \ a.s..
$$

By *The Martingale Convergence Theorem*, we get $\lim_{n \to +\infty} r(\theta_n) < +\infty$ $a.s.$ That is, for any $0 < \alpha < 1$, we have

$$
f(\theta_n) = O(\|\theta_n\|^{-\alpha}) \ a.s..
$$

By the arbitrariness of $\alpha$, we know the $O$ can be written as $o$, which indicates

$$
\min_{1 \leq k \leq n} \left\| \frac{\theta_k}{\|\theta_k\|} - \frac{\theta^*}{\|\theta^*\|} \right\| = o\left( \min_{1 \leq k \leq n} \|\theta_k\|^{\frac{-\alpha}{2}} \right) = o\left( \ln^{-\frac{\alpha}{2}} \min_{1 \leq k \leq n} g(\theta_k) \right) \ (\forall\, 0 < \alpha < 1) \ a.s..
$$

Through Lemma B.4 and Lemma B.8, we know

$$
\min_{1 \leq k \leq n} g(\theta_k) \leq \sqrt{\frac{1}{k_1^2} \min_{1 \leq k \leq n} \{\|\nabla g(\theta_k)\|^2\}} \leq \sqrt{\frac{\sqrt{\hat{K}n}}{nk_1} \sum_{k=2}^{+\infty} \frac{\|\nabla g(\theta_k)\|^2}{\sqrt{S_{k-1}}}} = O(n^{-\frac{1}{4}}) \ a.s..
$$

As a result, we obtain

$$
\min_{1 \leq k \leq n} \left\| \frac{\theta_k}{\|\theta_k\|} - \frac{\theta^*}{\|\theta^*\|} \right\| = o\left( \ln^{-\frac{1-\epsilon}{2}} n \right) \ (\forall\, \epsilon > 0) \ a.s..
$$

This completes the proof. $\qquad\square$

## C.11 GENERALIZING TO TIGHT EXPONENTIAL-TAIL LOSS

We show how our technique remains applicable when dealing with tight exponential-tail loss. This demonstration reinforces the enduring relevance of Lemma B.8 under the exponential-tail loss setting.

Before we give the main steps, we give the definition of 'tight exponential tail loss', by adopting the definition in Gunasekar et al. (2018).

**Definition 1.** *Consider a general classification problem, i.e., $\mathcal{L}(w) = \sum_{n=1}^{N} l(y_n w^T x_n)$, where $\{x_n, y_n\}_{n=N}^{2}$ is a dataset and labels $y_n \in \{-1, 1\}$ . Suppose the gradient $l'(u)$ satisfies:*

$$\forall\, u > u_+ :\; l'(u) \leq c(1 + e^{-u_+ + u})e^{-au},$$
$$\forall\, u > u_- :\; l'(u) \geq c(1 - e^{-u_- - u})e^{-au}, \tag{76}$$

*where $u_+ > 0$, $u_- > 0$, $a > 0$ are three constants, and $\lim_{u \to +\infty} l(u) = \lim_{u \to +\infty} l'(u) = 0$ , $l'(u) < 0$ , then we call $\mathcal{L}$ a tight exponential-tail loss (refer to Assumption 2, Definition 2, and Assumption 3 in Gunasekar et al. (2018) for reference).*

By solving the corresponding differential equation, we derive

$$\forall\, u > u_+ :\; l(u) \leq c' e^{-au}, \;\; \forall\, u > u_- :\; l(u) \geq c'' e^{-au}. \tag{77}$$

By combining the aforementioned inequalities and $\nabla \mathcal{L}(w) = \sum_{n=1}^{N} l'(w(t)^T x_n) y_n x_n$, we can derive a similar result to Lemma B.8 under the 'tight exponential tail loss' setting. We will then clarify it.

The left side of Lemma B.8 is derived as follows. Since $\{x_n, y_n\}$ is a linearly separable dataset, it has a maximum margin vector $\omega^*$. The margin vector $\omega^*$ satisfies the separation of $y_n = 1,\; (\omega^*)^T x_n > 0$ and $y_n = -1,\; (\omega^*)^T x_n < 0$. In addition, it has a lower bound $r := \min_n\{\|(\omega^*)^T x_n\|\}$ . Then, we acquire

$$\|\nabla \mathcal{L}(\theta)\| = \left\| \sum_{n=1}^{N} l'(y_n \omega(t)^T x_n) y_n x_n \right\| \geq \left| (\omega^*)^T \sum_{n=1}^{N} l'(y_n \omega(t)^T x_n) y_n x_n \right|$$

$$= \| \sum_{n=1}^{N} l'(y_n \omega(t)^T x_n) y_n (\omega^*)^T x_n \|.$$

Due to the conditions $y_n = 1,\; (\omega^*)^T x_n > 0$ and $y_n = -1,\; (\omega^*)^T x_n < 0$, all the signs of $\{y_n (\omega^*)^T x_n\}$ are the same. Then, we get

$$\|\nabla \mathcal{L}(\theta)\| \geq \left\| \sum_{n=1}^{N} l'(y_n \omega(t)^T x_n) y_n \omega^{*T} x_n \right\| = \sum_{n=1}^{N} \|l'(y_n \omega(t)^T x_n)\| \cdot \|y_n \omega^{*T} x_n\|$$

$$\geq r \sum_{n=1}^{N} \|l'(y_n \omega(t)^T x_n)\|.$$

For the right side of Lemma B.8, we use the triangle inequality, i.e.,

$$\|\nabla \mathcal{L}(w)\| = \left\| \sum_{n=1}^{N} l'(w(t)^T x_n) y_n x_n \right\| \leq \hat{r} \sum_{n=1}^{N} |l'(w(t)^T x_n)|,$$

where $\hat{r} > 0$ is a given scalar.

According to Equation (77) and Equation (76), we can observe that Lemma B.8 still holds.

