# OpenReview forum: "The Implicit Bias of Stochastic AdaGrad-Norm on Separable Data"
_ICLR.cc/2025/Conference — Submitted to ICLR 2025_

### Official Review · Reviewer_7jcj · 2024-10-28

**Soundness:** 3
**Presentation:** 3
**Contribution:** 3
**Rating:** 6
**Confidence:** 2

**Summary:**

This paper analyses the Adagrad-Norm algorithm on linearly separable data. The goal is study its implicit bias; i.e., the $L^2$ margin of its iterates. In particular, Adagrad-Norm is run on the cross-entropy loss. Under a boundedness assumption on the stochastic gradient norms (Assumption 3.1), the authors show that Adagrad-Norm converges to the max-margin solution almost surely (Theorem 4.2) and additionally characterize the rate of convergence (Theorem 4.3). They also show that under mini-batch stochastic gradients, the assumption holds and, thus, so do the results (Corollaries 4.1 and 4.2). The paper also conducts a small experiment to show that gradient descent and Adagrad-Norm both converge to the max-margin, while the algorithm Adagrad-Diagonal does not.

**Strengths:**

- The paper is clear and organized.

- Understanding the implicit bias of algorithms is important and the paper contributes in this direction by not only showing the convergence to the max-margin but also the rate of convergence.

- Rather than restricting to mini-batch stochastic gradients, the analysis holds more generally as long as Assumption 3.1 is satisfied.

**Weaknesses:**

- On page 8, when comparing the convergence to GD, I think it would be beneficial to explicitly mention what its rate is.

- The small dataset used for empirical validation seems limited in scope. It would be interesting to see how robust the results are for larger and more complex data.

- The paper would benefit from more concrete examples where Assumption 3.1 holds, as it is the main ingredient in the theoretical results.

**Questions:**

- Is there a reason why only 8 data points were used? Are results similar when there is more data? How about for higher-dimensional data?

- Given that the theoretical results hinge on linear separability, how applicable are these findings to real-world problems where data may not be perfectly separable?

---

### Official Review · Reviewer_VKqQ · 2024-11-02

**Soundness:** 2
**Presentation:** 2
**Contribution:** 2
**Rating:** 3
**Confidence:** 4

**Summary:**

This work studies the convergence of stochastic AdaGrad-Norm in the presence of linearly separable datasets. In particular, the authors want to explore if this method can converge to the max-margin solution, and hence demonstrate its implicit bias property to achieve good generalization without using regularization. A convergence rate result is also provided. Some techniques like divide-and-conquer based on the distance to the stationary point are proposed, which seem to be different from previous convergence analysis for AdaGrad-Norm type of algorithms. A small simulation is provided to validate the theory.

**Strengths:**

1.	Studying the convergence and implicit bias of stochastic adaptive algorithms has not been studied yet. This work provides the first attempt to close the gap.
2. Technically, the analysis seems not trival compared to previous convergence analysis of AdaGrad-Norm.

**Weaknesses:**

1.	The paper is not written well. There are a couple of places to be revised. First, the definition of almost surely (a.s.) is not defined. From 163-166, it is said ‘e’ is the margin vector of the data set $(x_i,y_i)$. How is it defined? From 210-215, it is a little bit hard to understand why the policy gradients under Markiv sampling becomes to the assumption made in this paper. Please elaborate. Some concrete examples might be helpful here. From line 225-239, the explanations are not clear. For example, what is the relationship between equation (4) and the goal to prove the difference between the gradient norms at $\theta_{n+1}$ and $\theta_{n}$ to be small. In line 269, why $x_{i_n}$ is the closest one to $\theta_n$? In line 305, ‘refer’ should be ‘refer to’.
2.	In step 5 of the proof sketch, could you please elaborate how to derive this recursion.
3.	I understand that the main contribution of this work is to prove the implicit bias of stochastic AdaGrad-Norm in terms of the max-margin direction. However, the authors should mention more about why it is important to study the implicit bias. It would be more interesting to me if the authors can show some extra properties of this adaptive methods over non-adaptive methods when converging to the max-margin direction.
4.	The proof may need double check. In (5), the first term (denoted by -A, for example) at the right side has a negative sign, i.e., Left < -A + something. Then, in line 302-303, it shows that A < E(Hn), which does not give the inequality Left < -E(Hn) because of the negative sign. I suggest a more careful check needs to be done for the entire proof. Given the current form, I am not convinced by the current version.
5.	More experiments, e.g., in real datasets, should be provided.

Overall, this work has big room to improve, in terms of writing, clarification, proof, and experiments. I am lean on the negative side.

**Questions:**

Please refer to the questions in the weakness part.

---

### Official Review · Reviewer_WoNp · 2024-11-03

**Soundness:** 1
**Presentation:** 1
**Contribution:** 1
**Rating:** 3
**Confidence:** 3

**Summary:**

This paper investigates the implicit bias of the stochastic AdaGrad-Norm algorithm in the binary classification setup with logistic regression loss for linearly separable data. The authors provide the almost sure convergence result to the $\mathcal{L}^2$ max-margin solution, together with a quantitative convergence rate. This demonstrates the implicit regularization effect of stochastic Adagrad-Norm towards the max-margin solution.

**Strengths:**

Implicit regularization is a topic of interest in optimization theory, and the paper studies the popular Adagrad algorithm. It provides a setup where in-depth technical analysis can be performed.

**Weaknesses:**

The most significant issue is that I cannot trust the correctness of these results. I detected a number of technical errors and unclarity in the proof of Lemma B.4, which is the very first theoretical analysis that the authors provide, and is the basis of the proof of other theoretical results. I will list the details in the Questions section.

There are two possibilities. The first is that the authors' analysis is actually incorrect, and the second is that the authors' theory is correct but is very poorly presented with numerous typos, calculation misses, and unclear explanations. Even if the optimistic second possibility is the case, I vote for rejection of the paper because its current quality of presentation is poor and far away from the level that is adequate for publication. At the most technical level, the proofs are very sloppy and difficult to read through, and includes too many major/minor errors causing doubts on the soundness of the theoretical statements. Additionally, there are plenty of sentence-level grammar issues and parts whose meaning are unclear. At a higher level, the organization of the main paper features a large volume of technical information that contains notations that were never defined within the main text and hence is not comprehensible unless one goes through the appendix. In pg. 5, the paper declares through the paragraph title that it will illustrate the "Intuition of the theorem", but at least to me, only little proportion of the subsequent materials was informative.

Besides the above issues, there is a weakness that the paper considers a specific setup and it is questionable whether their theory and techniques (if are correct) can be generalized to broader problem classes.

**Questions:**

List of (serious) technical issues regarding the proof of Lemma B.4
- In derivation of equation (10), the authors seem to combine the previous equations. I think the $\alpha_0^3$ factor is missing from the second line, and more importantly, it seems like the term $\frac{\alpha_0}{2\hat{K}^2} \mathbf{1}_{\mathcal{B}_n} \frac{||\nabla g(\theta_n)||^2 ||\nabla g(\theta_n, \xi_n)||^2}{\sqrt{S_n}}$ has suddenly disappeared.
- In the lines 732-734, it seems that the authors have upper-bounded the term $-\mathbb{E} [ (1_{B_n} - 1_{B_n} 1_{B_{n+1}}) g(\theta_{n+1}) ]$ by $\min ( \delta_0, \delta_1 ) \cdot \mathbb{E}[ (1_{B_n} - 1_{B_n} 1_{B_{n+1}}) ]$, but how can this be true? The event $B_n$ does not imply that $g(\theta_{n+1}) \le \delta_2 = \min ( \delta_0, \delta_1 )$. Also, the second term has been bounded as $-\mathbb{E} [ (1_{B_n} 1_{B_{n+1}} - 1_{B_{n+1}}) g(\theta_{n+1}) ] \le \min ( \delta_0, \delta_1 ) \cdot \mathbb{E} (1_{B_n} 1_{B_{n+1}} - 1_{B_{n+1}})$, but I believe that the sign of the right hand side should be flipped, as it is currently nonpositive.
- I do not understand how lines 740-745 implies (12). There is a logical jump here, and I even do not consider the lines 740-745 reliable due to the aforementioned issues.
- In lines 758-762, I think the authors upper-bounded the inner product term $-1_{B_n}^{(-)} \frac{\alpha_0 \nabla g(\theta_n)^T \nabla g(\theta_n, \xi_n)}{\sqrt{S_n}}$ by  $-1_{B_n}^{(-)} \frac{\alpha_0}{2} ( \frac{1}{M+1} \frac{||\nabla g(\theta_n, \xi_n)||^2}{\sqrt{S_n}} + (M+1) \sqrt{||\nabla g(\theta_n)||^2}{\sqrt{S_n}})$ using Young's inequality, but the direction of the inequality is incorrect. First I guessed they might have mistakenly put the minus sign on the right hand side, but then I do not understand how the next inequality (line 767) is derived.

Below, I pinpoint the some writing issues including unclear sentences, grammar/vocabulary issues, missing/incorrect notations and typos, and other problems compromising the paper's flow and readability. Note that this is not even a comprehensive list, and the paper would require a lot of polishing with respect to the below list and more.
- In line 27, "optimums" should become "optima".
- In line 39, "varient" should become "variant".
- In line 55-56, "The $L^2$ max-margin solution $\theta^* / ||\theta^*||$ as the vectors ..." is grammatically incorrect. Something like "is defined" should be inserted in the middle. Also, some discussion on the uniqueness of (2) might be needed here.
- In lines 77-78, the phrase "are based on the situation that mini-batch stochastic gradient holds" is unclear, both in terms of the grammar and meaning. Please be precise with technical terminologies at least. I think "mini-batch stochastic gradient" isn't something that can "hold" in any sense.
- In lines 84-86, the sentence "For example, ..., bypassed by considering the dynamic system in different segments." is entirely unclear to me. I think there is no point of including this discussion if the authors can't clearly convey the high-level intuition behind what the challenges are to the readers who are not already familiar with the detailed proof techniques they have.
- In line 96, the authors mention some prior work on AdaGrad-Diagonal (which they haven't defined at this point) that are seemingly contradictory to the paper's main results, and this was confusing during the first pass. Please further clarify how AdaGrad-Norm and AdaGrad-Diagonal are different, and why this isn't a contradiction.
- In line 138, the concept of filtration appears suddenly. Does it belong to this location?
- In lines 153-154, what did the authors precisely mean by "Since the choice of logistic regression does not affect the validity of our analysis"?
- In line 186, "there is" seems to be inserted mistakenly.
- In line 204-206, the notation $k_2$ appears suddenly without clarifying what it is. Although the authors mention Lemma B.8, I think they should explicitly mention that $k_2$ is a constant that is used for bounding the gradient of $g$ in terms of the function value.
- In line 211, what does it mean that policy learning "a classification problem in the policy space"?
- In lines 214-215 the authors say "since the rewards ... in the reinforcement learning setting are bounded, it can be shown that the stochastic gradients are almost surely bounded", which seems to be an overly strong statement. Are they claiming that this true within arbitrary RL setups? If not, precisely which cases are they referring to?
- In lines 228, 234, "inequity" were used in place of "inequality".
- In line 256, how is the law of large numbers (LLN) related to approximating $-\nabla g(\theta_n, \xi_n)$ by $-\nabla g(\theta_n)$ as $n\to \infty$? Aren't the gradient noises independent at every iteration and LLN has nothing to do with them?
- In the equation spanning lines 261-267, the undefined notation like $f_{x_i}$ is used, and it is not clearly specified what $A_n$ and $B_n$ precisely are. In fact, I believe that having this equation does not align with the purpose of illustrating the intuition of the proof because there is no clear explanation of what this expansion means, and as a reader, I have not gained any idea of what the terms $A_n, B_n$ are and where they come from.
- In line 271, "That concludes $A_n$ is the dominate term" seems grammatically incorrect.
- In line 282, what does it precisely mean by "which tends to $||\cdots||$ as $||\theta|| \to + \infty$"? I think the authors could explain more on how $f(\theta_n) \to 0$ a.s. implies their desired implicit bias.
- In line 305, "where the definition of ... can refer Equation (38) in ..." seems grammatically incorrect.
- Isn't equation (6) completely redundant with lines 261-267?
- I felt very demotivated to carefully read through Steps 1~5 of Section 4, where the authors make use of numerous notations including $T_n, f_{x_i}, C_n, D_n, \mathcal{L}$ and $s'$ whose meaning cannot be fully understood at the moment of reading through the main text, which lacks a sufficiently motivating explanation. I speculate that this can happen to many readers. I recommend the authors to rewrite this section with more careful consideration of which essential ideas and intuition they trying to convey, and how this can be done without bombarding the readers with unfamiliar mathematical terms.
- In Lemma B.3, $\epsilon$ has been used without being specified what it is.
- In line 635 (the first line of the equation), $\nabla f(\theta_n))^T$ has an extra $)$.
- In lines 674-675, I think it is authors' responsibility to show through calculations that $||\nabla^2 g(\theta)|| = \Theta(||\nabla g(\theta)||)$. Furthermore, the meaning of $\Theta$ notation is not only that $||\nabla^2 g(\theta)|| \le \tilde{d}_0 ||\nabla g(\theta)||$, but the reverse inequality should hold true with some constant other than $\tilde{d}_0$.
- In lines 675-676, "That means existing ..." seems grammatically incorrect.
- In line 695, I think they have a typo: $\hat{\delta}_1$ in place of $\tilde{\delta}_1$.

---

> ### Author Response · Authors · 2024-11-17
>
> Thank you for your comments. However, the four critical errors you pointed out are all based on misunderstandings. Below, we will respond specifically to these four issues.
>
> For the first question, we do not miss the term
>
> $$\frac{\alpha_0}{2\hat{K^2}}\textbf{1}_{\mathcal{B}_{n}}\frac{\\|\nabla g(\theta_{n})\\|^{2}\\|\nabla g(\theta_{n},\xi_{n})\\|^{2}}{\sqrt{S_n}},$$
>
> as it is offset by the preceding term
>
> $$-\frac{\alpha_{0}}{2}\textbf{1}_{\mathcal{B}_{n}}\frac{\|\nabla g(\theta_{n})\|^{2}}{\sqrt{S_{n-1}}},$$
>
> since, when the event $$\mathcal{B}_{n}$ occurs, it is evident that $\|\nabla g(\theta_{n},\xi_{n})\|^{2}<\hat{K}^{2}$$.
>
>
>
> For the second question, we are not mistaken. Note that this is not merely the indicator function $\textbf{1}_{\mathcal{B}_{n}}$, but rather
>
> $$\textbf{1}_{\mathcal{B}_{n}} - \textbf{1}_{\mathcal{B}_{n}}\textbf{1}_{\mathcal{B}_{n+1}},$$
>
> which can be classified as
>
> $$\textbf{1}_{\mathcal{B}_{n}/\mathcal{B}_{n+1}}.$$
>
> Therefore, the corresponding inequality in the latter part follows naturally. Similarly, the second case can be classified as
>
> $$\textbf{1}_{\mathcal{B}_{n+1}/\mathcal{B}_{n}}.$$
>
> These are standard mathematical techniques commonly used in such contexts.
>
> As for the third question, this is truly self-evident. Simply sum the recurrence relation above over the index \( n \) from 1 to infinity, then rearrange the terms, and Equation (12) immediately follows. Honestly, if this isn’t obvious, I really don’t know what qualifies as obvious.
>
> Regarding the last question, you are mistaken again. Here, we did not use the Young's inequality; instead, we employed the following simple algebraic transformation:
>
> $$
> -\textbf{1}_{\mathcal{B}_{n}}^{(-)}\frac{\alpha_0\nabla g(\theta_{n})^{T}\nabla g(\theta_{n},\xi_{n})}{\sqrt{S_{n}}} = -\textbf{1}_{\mathcal{B}_{n}}^{(-)}\frac{\alpha_0}{2} \Bigg(\frac{1}{M+1}\frac{\big\|\nabla g(\theta_{n},\xi_{n})\big\|^{2}}{\sqrt{S_{n}}} + (M+1)\frac{\big\|\nabla g(\theta_{n})\big\|^{2}}{\sqrt{S_{n}}}\Bigg)
> + \textbf{1}_{\mathcal{B}_{n}}^{(-)}\frac{\alpha_0}{2}\frac{1}{\sqrt{S_{n}}}\Big\|\frac{1}{\sqrt{M+1}}\nabla g(\theta_{n},\xi_{n}) - \sqrt{M+1}\nabla g(\theta_{n})\Big\|^{2}.
> $$
>
> Then, we bound it as:
>
> $$
> \leq -\textbf{1}_{\mathcal{B}_{n}}^{(-)}\frac{\alpha_0}{2} \Bigg(\frac{1}{M+1}\frac{\big\|\nabla g(\theta_{n},\xi_{n})\big\|^{2}}{\sqrt{S_{n}}} + (M+1)\frac{\big\|\nabla g(\theta_{n})\big\|^{2}}{\sqrt{S_{n}}}\Bigg)
> + \textbf{1}_{\mathcal{B}_{n}}^{(-)}\frac{\alpha_0}{2}\frac{1}{\sqrt{S_{n-1}}}\Big\|\frac{1}{\sqrt{M+1}}\nabla g(\theta_{n},\xi_{n}) - \sqrt{M+1}\nabla g(\theta_{n})\Big\|^{2}.
> $$
>
> This approach uses a straightforward splitting and regrouping of terms, not Young's inequality

---

> ### Comment · Reviewer_WoNp · 2024-11-19
>
> I thank the authors for the response. However, I would like to mention once again that, as I stated in the initial review, **even if the authors' analyses are correct in the end, it is authors’ responsibility to make all arguments as clear as possible and present it in a way that readers can comprehend everything easily.**
> I mentioned some technical issues within the proofs merely as **examples** showing how bumpy it felt, as a reader, to go through the analysis in general, and similar types of technical sloppiness seem to continue in the remaining proof of Lemma B.4 and in proofs of other theorems as well.
> Therefore, clarifying the specific examples I mentioned at this stage is not sufficient for changing this evaluation.
> As someone with experience of writing theory papers, I strongly believe that such polishing of arguments should have been done in the first place, throughout the whole paper.
> The level of clarity achieved by the authors’ original arguments combined with additional explanations from the rebuttal is the minimum line that I think what qualifies as an acceptable theory paper.
>
> By the way, one of the additional arguments from the response is yet unsatisfactory.
> In the response to the third point I mentioned, the authors said “if this isn’t obvious, I really don’t know what qualifies as obvious”.
> But it is not that I asked the question because I don’t know what telescoping is.
> To claim (12), I think the authors should have discussed why the infinite sum of expectations of $(\hat{c}\hat{K} + 2\hat{d}_0^2 \hat{K}^2) ||\nabla g(\theta_n, \xi_n)||^2 / S_n^{3/2}$ is finite (I thought this may be an application of Lemma B.3, but it's different because the numerator has $\xi_n$ dependency). At least, that’s what I certainly would have done if I were writing down the same proof.
> Maybe showing that would be straightforward. I’m not claiming that this is something technically so important that it determines the value of the paper. I’m saying that I do not agree with the authors’ attitude of leaving such details to the reader and simply marking it as “obvious”.
>
> Note that the paper also still has a number of unresolved general writing issues, independent of the above discussion.

---

> > ### Author Response · Authors · 2024-11-20
> >
> > $$\sum_{n=1}^{+\infty}\frac{\\|\nabla g(\theta_{n},\xi_{n})\\|^{2}}{S_{n}^{\frac{3}{2}}}<\int_{S_{1}}^{+\infty}\frac{\text{d}x}{x^{\frac{3}{2}}}<+\infty.$$
> > This is a very common scaling technique for series and integrals, probably something learned in the first year of undergraduate studies. I really believe this is something every competent reviewer should know.

---

> > > ### Comment · Reviewer_WoNp · 2024-11-20
> > >
> > > I respectfully disagree. I believe the simplicity of the math consisting an argument does not necessarily indicate that the clarity can be preserved whenever it is omitted without a proper explanation.

---

> > > > ### Author Response · Authors · 2024-11-20
> > > >
> > > > Since this proof is already quite lengthy, we have omitted some parts that we believe are relatively obvious.

---

### Official Review · Reviewer_GjCV · 2024-11-04

**Soundness:** 3
**Presentation:** 2
**Contribution:** 2
**Rating:** 6
**Confidence:** 3

**Summary:**

This paper investigates the stochastic AdaGrad-Norm method, with a broad range of sampling noise, applied to linearly separable classification problems. Specifically, it proves that the method exhibits an implicit bias, converging almost surely to the L2 max-margin solution without explicit regularization.

**Strengths:**

Examining the implicit bias of optimization methods is an important research direction for understanding the statistical efficiency of machine learning models. This work uncovers the implicit bias of AdaGrad-Norm along with its convergence rate.

**Weaknesses:**

- Given the existing literature on the implicit bias of optimization methods, the primary concern is the significance of the results presented. For instance, the classic result by [Z. Ji and M. Telgarsky] demonstrates a convergence rate $\log\log n/\log n$ of GD to the L2-margin solution, which is faster than the rate shown in this submission. Moreover, [C. Zhang, D. Zou, and Y. Cao] have shown much faster rates for Adam converging to the L-infinity margin solution. This submission also lacks citations to these papers and other relevant works:

[Z. Ji and M. Telgarsky] The implicit bias of gradient descent on nonseparable data, COLT 2019.

[C. Zhang, D. Zou, and Y. Cao] The Implicit Bias of Adam on Separable Data. 2024.

[S. Xie and Z. Li] Implicit Bias of AdamW: l_\infty-Norm Constrained Optimization. ICML 2024

[M. Nacson, N. Srebro, and D. Soudry] Stochastic gradient descent on separable data: Exact convergence with a fixed learning rate.  AISTATS 2019.

- Since AdaGrad-Norm has the same implicit bias as GD, the advantages of using AdaGrad-Norm over GD are unclear.

- The bounded noise assumption, while common, is somewhat restrictive in stochastic optimization literature. There have been several efforts to extend these noise conditions:

[A. Khaled and P. Richt´arik]. Better theory for sgd in the nonconvex world. TMLR 2023.
[R. Gower, O. Sebbouh, and N. Loizou] Sgd for structured nonconvex functions: Learning rates, minibatching and interpolation. AISTATS 2021.

**Questions:**

Can you explain the theoretical advantage of AdaGrad-Norm compared to GD? Currently, it is unclear as both methods exhibit the same implicit bias.

---

### Meta-Review · Area_Chair_eaTQ · 2024-12-19

**Metareview:**

The paper explores the implicit bias of stochastic AdaGrad-Norm on linearly separable data, showing convergence to the $L_2$ max-margin solution. Reviewers acknowledged the relevance of the topic and the paper's focus on adaptive gradient methods. However, concerns were raised about the clarity of the presentation, the rigor of the theoretical analysis, and the novelty of the results compared to existing work. Despite attempts to clarify during the rebuttal, these issues remained unresolved. I recommend rejection, as the work requires significant improvements to meet the bar for acceptance.

**Additional Comments On Reviewer Discussion:**

During the discussion, reviewers raised concerns about the clarity of the presentation, the rigor of the theoretical arguments, and the novelty of the results compared to prior work. The authors responded by attempting to clarify some of the derivations and arguments. However, the reviewers felt that the responses did not fully resolve the issues, particularly regarding the lack of significant theoretical or empirical advances. After considering the discussion and rebuttal, I concluded that the concerns remained substantial, leading to the decision to recommend rejection.

---

### Decision · Program_Chairs · 2025-01-22

Reject